# Thompson Sampling for Non-Stationary Bandit Problems

## Abstract

Non-stationary multi-armed bandit (MAB) problems have recently attracted extensive attention. We focus on the abruptly changing scenario where reward distributions remain constant for a certain period and change at unknown time steps. Although Thompson Sampling (TS) has shown empirical success in non-stationary settings, there is currently no regret bound analysis for TS with Gaussian priors. To address this, we propose two algorithms, discounted TS and sliding-window TS, designed for sub-Gaussian reward distributions. For these algorithms, we establish an upper bound for the expected regret by bounding the expected number of times a suboptimal arm is played. We show that the regret order of both algorithms is $\tilde{O}(\sqrt{TB_T})$, where $T$ is the time horizon, $B_T$ is the number of breakpoints. This upper bound matches the lower bound for abruptly changing problems up to a logarithmic factor. Empirical comparisons with other non-stationary bandit algorithms highlight the competitive performance of our proposed methods.

## 1 Introduction

MAB is a classic sequential decision problem. At each time step, the learner selects an arm from a finite set of arms (also known as actions) based on its past observations, and she only observes the reward of the chosen action. The learner's goal is to maximize its expected cumulative reward or minimize the regret incurred during the learning process. The regret is defined as the difference between the expected reward of the optimal arm and the expected reward achieved by the MAB algorithm.

MAB has found practical use in various scenarios, with one of the earliest applications being the diagnosis and treatment experiments proposed by Robbins (1952). In this experiment, each patient's treatment plan corresponds to an arm in the MAB problem, and the goal is to minimize the patient's health loss by making optimal treatment decisions. Recently, MAB has gained wide-ranging applicability. For example, MAB algorithms have been used in online recommendation systems to improve user experiences and increase engagement (Li et al., 2011; Bouneffouf et al., 2012; Li et al., 2016). Similarly, MAB has been employed in online advertising campaigns to optimize the allocation of resources and maximize the effectiveness of ad placements (Schwartz et al., 2017). While the standard MAB model assumes fixed reward distributions, real-world scenarios often involve changing distributions over time. For instance, in online recommendation systems, the collected data gradually becomes outdated, and user preferences are likely to evolve (Wu et al., 2018). This dynamic nature necessitates the development of algorithms that can adapt to these changes, leading to the exploration of non-stationary MAB problems.

In recent years, there has been much research on non-stationary multi-armed bandit problems. These methods can be roughly divided into two categories: they either detect changes in the reward distribution using change-point detection algorithms (Liu et al., 2018; Cao et al., 2019; Auer et al., 2019; Chen et al., 2019; Besson et al., 2022), or they passively reduce the effect of past observations (Garivier & Moulines, 2011; Raj & Kalyani, 2017; Trovo et al., 2020; Baudry et al., 2021). The former method needs to make some assumptions about the change of arms distribution to ensure the effectiveness of the change-point detection algorithm. For instance, (Liu et al., 2018; Cao et al., 2019) require a lower bound on the amplitude of change of each arm's expected rewards. The latter requires fewer assumptions about the characteristics of the change. They often use a sliding window or discount factor to forget past information to adapt to the change of arms distribution.

These methods all provide the theoretical guarantees for regret upper bounds. However, the known Thompson sampling, have received little theoretical analysis of regret in non-stationary MAB problems, despite the fact TS algorithms often have superior or comparable performance to frequentist algorithms in most non-stationary scenarios. Raj & Kalyani (2017) have studied the discounted Thompson sampling with Beta priors. However, they only derive the probability of picking a suboptimal arm for the simple case of a two-armed bandit. To the best of our knowledge, only sliding-window Thompson sampling with Beta priors (Trovo et al., 2020) provides the regret upper bounds. However, their proof is incorrect with a wrong application of a well-known result (Lemma A.3). We analyze their mistakes in detail and provide a counterexample in Appendix C.

There are two main challenges in analyzing Thompson sampling algorithm in non-stationary setting. The first challenge is that the DS-TS algorithm cannot **fully forget previous information** and the second is the **under-estimation of the optimal arm**. In non-stationary setting, solving these problems is highly challenging due to the changing reward distribution. We define a UCB-like function serving as the upper confidence bound to tackle the first challenge, as detailed in Lemma 5.1 and Lemma 5.2. Along with the defined function, we employ a new regret decomposition to bound the regret comes from the under-estimation of the optimal arm, as presented in the proof of Lemma 5.3. We provide details about these challenges and their solutions in the theoretical analysis section (Section 5).

Our contributions are as follows: we propose discounted TS (DS-TS) and sliding-window TS (SW-TS) with Gaussian priors for abruptly changing settings. We adopt a unified method to analyze the regret upper bound for both algorithms. The theoretical analysis results show that their regret upper bounds are of order $\tilde{O}(\sqrt{TB_T})$, where $T$ is the number of time steps, $B_T$ is the number of breakpoints. This regret bound matches the $\Omega(\sqrt{T})$ lower bound proven by Garivier & Moulines (2011) in an order sense. We also verify the algorithms in various environmental settings with Gaussian and Bernoulli rewards, and both DS-TS and SW-TS achieve competitive performance.

## 2 Related Works

Many works are based on the idea of forgetting past observations. Discounted UCB (DS-UCB) (Kocsis & Szepesvári, 2006; Garivier & Moulines, 2011) uses a discounted factor to average the past rewards. In order to achieve the purpose of forgetting information, the weight of the early reward is smaller. Garivier & Moulines (2011) also propose the sliding-window UCB (SW-UCB) by only using a few recent rewards to compute the UCB index. They calculate the regret upper bound for DS-UCB and SW-UCB as $\tilde{O}(\sqrt{TB_T})$. EXP3.S, as proposed in (Auer et al., 2002), has been shown to achieve the regret upper bound by $\tilde{O}(\sqrt{TB_T})$. Under the assumption that the total variation of the expected rewards over the time horizon is bounded by a budget $V_T$, Besbes et al. (2014) introduce REXP3 with regret $\tilde{O}(T^{2/3})$. Combes & Proutiere (2014) propose the SW-OSUB algorithm, specifically for the case of smoothly changing with an upper bound of $\tilde{O}(\sigma^{1/4}T)$, where $\sigma$ is the Lipschitz constant of the evolve process. Raj & Kalyani (2017) propose the discounted Thompson sampling for Bernoulli priors without providing the regret upper bound. They only calculate the probability of picking a sub-optimal arm for the simple case of a two-armed bandit. Trovo et al. (2020) propose the sliding-window Thompson sampling algorithm with regret $\tilde{O}(T^{\frac{1+\alpha}{2}})$ for abruptly changing settings and $\tilde{O}(T^{\beta})$ for smoothly changing settngs. Baudry et al. (2021) propose a novel algorithm named Sliding Window Last Block Subsampling Duelling Algorithm (SW-LB-SDA) with regret $\tilde{O}(\sqrt{TB_T})$. They only assume that the reward distributions belong to the same one-parameter exponential family for all arms during each stationary phase.

There are also many works that exploit techniques from the field of change detection to deal with reward distributions varying over time. Mellor & Shapiro (2013) combine a Bayesian change point mechanism and Thompson sampling strategy to tackle the non-stationary problem. Their algorithm can detect global switching and per-arm switching. Liu et al. (2018) propose a change-detection framework that combines UCB and a change-detection algorithm named CUSUM. They obtain an upper bound for the average detection delay and a lower bound for the average time between false alarms. Cao et al. (2019) propose M-UCB, which is similar to CUSUM but uses another simpler change-detection algorithm. M-UCB and CUSUM are nearly optimal, their regret bounds are $\tilde{O}(\sqrt{TB_T})$.

Recently, there are also some works deriving regret bounds without knowing the number of changes. For example, Auer et al. (2019) propose an algorithm called ADSWITCH with optimal regret bound $\tilde{O}(\sqrt{B_T T})$. Suk & Kpotufe (2022) improve the work (Auer et al., 2019) so that the obtained regret bound is smaller than $\tilde{O}(\sqrt{ST})$, where $S$ only counts the best arms switches.

## 3 Problem Formulation

Assume that the non-stationary MAB problem has $K$ arms $\mathcal{A} := \{1, 2, ..., K\}$ with finite time horizon $T$. At each round $t$, the learner must select an arm $i_t \in \mathcal{A}$ and obtain the corresponding reward $X_t(i_t)$. The rewards are generated from $\sigma$-subGaussian distributions. The expectation of $X_t(i)$ is denoted as $\mu_t(i) = \mathbb{E}[X_t(i)]$. A policy $\pi$ is a function that selects arm $i_t$ to play at round $t$. Let $\mu_t(*) := \max_{i \in \{1, ..., K\}} \mu_t(i)$ denote the expected reward of the optimal arm $i_t^*$ at round $t$. Unlike the stationary MAB settings, where an arm is optimal all of the time (i.e. $\forall t \in \{1, ..., T\}, i_t^* = i^*$), while in the non-stationary settings, the optimal arms might change over time. The performance of a policy $\pi$ is measured in terms of cumulative expected regret:

$$R_T^\pi = \mathbb{E}\left[ \sum_{t=1}^{T} (\mu_t(*) - \mu_t(i_t)) \right], \tag{1}$$

where $\mathbb{E}[\cdot]$ is the expectation with respect to randomness of $\pi$. Let $\Delta_t(i) = \mu_t(*) - \mu_t(i)$ and let

$$k_T(i) = \sum_{t=1}^{T} \mathbb{1}\{i_t = i, i \neq i_t^*\}$$

denote the number of plays of arm $i$ when it is not the best arm until time $T$. When we analyze the upper bound of $R_T^\pi$, we can directly analyze $\mathbb{E}[k_T(i)]$ to get the upper bound of each arm.

**Abruptly Changing Setting** The abruptly changing setting is introduced by Garivier & Moulines (2011) for the first time. The number of breakpoints is denoted as $B_T = \sum_{t=1}^{T-1} \mathbb{1}\{\exists i \in \mathcal{A} : \mu_t(i) \neq \mu_{t+1}(i)\}$. Suppose the set of *breakpoints* is $\mathcal{B} = \{b_1, ..., b_{B_T}\}$ (we define $b_1 = 1$). At each breakpoint, the reward distribution changes for at least one arm. The rounds between two adjacent breakpoints are called *stationary phase*. Abruptly changing bandits pose a more challenging problem as the learner needs to balance exploration and exploitation within each stationary phase and during the changes between different phases. Trovo et al. (2020) makes assumption about the number of breakpoints to facilitate more generalized analysis, while we explicitly use $B_T$ to represent the number of breakpoints for analysis.

## 4 Algorithms

In this section, we propose the DS-TS and SW-TS with Gaussian priors for the non-stationary stochastic MAB problems. Different from Agrawal & Goyal (2013), we assume that the reward distribution follows a $\sigma$-subGaussian distribution rather than a bounded distribution. Assume that $X_1, ..., X_n$ are independently and identically distributed, following a $\sigma$-subGaussian distribution with mean $\mu$. Assume further that the prior distribution is a Gaussian distribution $\mu \sim \mathcal{N}(0, \sigma_0^2)$. The posterior distribution is also Gaussian distribution $\mathcal{N}(\mu_1, \sigma_1^2)$ where

$$\mu_1 = \sigma_1^2 \left( \frac{0}{\sigma_0^2} + \frac{\sum_{i=1}^{n} X_i}{\sigma^2} \right), \sigma_1^2 = \frac{1}{\frac{1}{\sigma_0^2} + \frac{n}{\sigma^2}}.$$

Let $\sigma_0 = +\infty$, we get the posterior distribution as $\mathcal{N}(\frac{1}{n} \sum_{i=1}^{n} X_i, \frac{\sigma^2}{n})$.

### 4.1 DS-TS

DS-TS uses a discount factor $\gamma$ ($0 < \gamma < 1$) to dynamically adjust the estimate of each arm's distribution. The key to our algorithm is to decrease the sampling variance of the selected arm while increasing the sampling variance of the unselected arms.

---

**Algorithm 1:** DS-TS

**Input:** discounted factor $\gamma$, $\hat{\mu}_1(i) = 0$, $\tilde{\mu}_1(i) = 0$, $N_t(\gamma, i) = 0$

1 **for** $t = 1, ..., T$ **do**
2     **for** $i = 1, .., K$ **do**
3         sample $\theta_t(i) \sim \mathcal{N}(\hat{\mu}_t(\gamma, i), \frac{4\sigma^2}{N_t(\gamma,i)})$
4     **end**
5     Pull arm $i_t = \arg\max_i \theta_t(i)$, observe reward $X_t(i_t)$;
6     **for** $i = 1, ..., K$ **do**
7         $\tilde{\mu}_{t+1}(\gamma, i) = \gamma\tilde{\mu}_t(\gamma, i) + \mathbb{1}\{i_t = i\}X_t(i)$
8         $N_{t+1}(\gamma, i) = \gamma N_t(\gamma, i) + \mathbb{1}\{i_t = i\}$
9         $\hat{\mu}_{t+1}(\gamma, i) = \frac{\tilde{\mu}_{t+1}(\gamma,i)}{N_{t+1}(\gamma,i)}$
10    **end**
11 **end**

---

Specifically, let

$$N_t(\gamma, i) = \sum_{j=1}^{t} \gamma^{t-j} \mathbb{1}\{i_j = i\}$$

denote the discounted number of plays of arm $i$ until time $t$. We use

$$\hat{\mu}_t(\gamma, i) = \frac{1}{N_t(\gamma, i)} \sum_{j=1}^{t} \gamma^{t-j} X_j(i) \mathbb{1}\{i_j = i\}$$

called discounted empirical average to estimate the expected rewards of arm $i$. In non-stationary settings, we use the discounted average and discounted number of plays instead of the true average and number of plays respectively. Therefore, the posterior distribution is $\mathcal{N}(\hat{\mu}_t(\gamma, i), \frac{\sigma^2}{N_t(\gamma,i)})$.

---

**Algorithm 2:** SW-TS

**Input:** sliding window $\tau$, $\hat{\mu}_1(i) = 0$, $\tilde{\mu}_1(i) = 0$, $N_t(\tau, i) = 0$

1 **for** $t = 1, ..., T$ **do**
2     **for** $i = 1, .., K$ **do**
3         sample $\theta_t(i) \sim \mathcal{N}(\hat{\mu}_t(\tau, i), \frac{4\sigma^2}{N_t(\tau,i)})$
4     **end**
5     Pull arm $i_t = \arg\max_i \theta_t(i)$, observe reward $X_t(i_t)$
6     **for** $i = 1, ..., K$ **do**
7         $N_{t+1}(\tau, i) = N_t(\tau, i) + \mathbb{1}\{i_t = i\} - \mathbb{1}\{i_{t-\tau} = i\}$
8         $\tilde{\mu}_{t+1}(\tau, i) = \tilde{\mu}_t(\tau, i) + \mathbb{1}\{i_t = i\}X_t(i) - \mathbb{1}\{i_{t-\tau} = i\}X_{t-\tau}(i)$
9         $\hat{\mu}_{t+1}(\tau, i) = \frac{\tilde{\mu}_{t+1}(\tau,i)}{N_{t+1}(\tau,i)}$
10    **end**
11 **end**

---

Algorithm 1 shows the pseudocode of DS-TS. Step 3 is the Thompson sampling. For each arm, we draw a random sample $\theta_t(i)$ from $\mathcal{N}(\hat{\mu}_t(\gamma, i), \frac{4\sigma^2}{N_t(\gamma,i)})$. We use $\frac{4\sigma^2}{N_t(\gamma,i)}$ as the posterior variance instead of $\frac{\sigma^2}{N_t(\gamma,i)}$, which helps the subsequent analysis. Then we select arm $i_t$ with the maximum sample value and obtain the reward $X_t(i_t)$ (Step 5). To avoid the time complexity going to $O(T^2)$, we introduce $\tilde{\mu}_t(\gamma, i) = \sum_{j=1}^{t} \gamma^{t-j} X_j(i)\mathbb{1}\{i_j = i\}$ to calculate $\hat{\mu}_t(\gamma, i)$ using an iterative method(Step 7-9).

If arm $i$ is selected at round $t$, the posterior distribution is updated as follows:

$$\hat{\mu}_{t+1}(\gamma, i) = \frac{\gamma\hat{\mu}_t(\gamma, i)N_t(\gamma, i) + X_t(i)}{\gamma N_t(\gamma, i) + 1} = \frac{\tilde{\mu}_{t+1}(\gamma, i)}{N_{t+1}(\gamma, i)}$$

If arm $i$ isn't selected at round $t$, the posterior distribution is updated as

$$\hat{\mu}_{t+1}(\gamma, i) = \frac{\tilde{\mu}_{t+1}(\gamma, i)}{N_{t+1}(\gamma, i)} = \frac{\gamma \tilde{\mu}_t(\gamma, i)}{\gamma N_t(\gamma, i)} = \hat{\mu}_t(\gamma, i)$$

i.e. the expectation of posterior distribution remains unchanged.

### 4.2 SW-TS

SW-TS uses a sliding window $\tau$ to adapt to changes in the reward distribution. Let

$$N_t(\tau, i) = \sum_{j=t-\tau+1}^{t} \mathbb{1}\{i_j = i\}, \hat{\mu}_t(\tau, i) = \frac{1}{N_t(\tau, i)} \sum_{j=t-\tau+1}^{t} X_j(i)\mathbb{1}\{i_j = i\}.$$

If $t < \tau$, the range of summation is from 1 to $t$. Similar to DS-TS, the posterior distribution is $\mathcal{N}(\hat{\mu}_t(\tau, i), \frac{4\sigma^2}{N_t(\tau, i)})$. Algorithm 2 shows the pseudocode of SW-TS. To avoid the time complexity going to $O(T^2)$, we introduce $\tilde{\mu}_t(\tau, i) = \sum_{j=t-\tau+1}^{t} X_j(i)\mathbb{1}\{i_j = i\}$ to update $\hat{\mu}_t(\tau, i)$.

### 4.3 Results

In this section, we give the regret upper bounds of DS-TS and SW-TS. Then we discuss how to take the values of the parameters so that these algorithms reach the optimal upper bound.

Recall that $\Delta_t(i) = \mu_t(*) - \mu_t(i)$. Let $\Delta_T(i) = \min\{\Delta_t(i) : t \leq T, i \neq i_t^*\}$, be the minimum difference between the expected reward of the best arm $i_t^*$ and the expected reward of arm $i$ in all time $T$ when the arm $i$ is not the best arm. Let $\Delta_{max}^T = \max\{\mu_{t_1}(i) - \mu_{t_2}(i) : t_1 \neq t_2, i \in [K]\}$ denote the maximum expected variation of arms.

**Theorem 4.1** (DS-TS). *Let $\gamma \in (0, 1)$ satisfying $\frac{\sigma^2}{\Delta_{max}^T}(1 - \gamma)^2 \log \frac{1}{1-\gamma} < 1$. For any suboptimal arm $i$,*

$$\mathbb{E}[k_T(i)] \leq B_T D(\gamma) + C_1(\gamma)L_1(\gamma)\gamma^{-\frac{1}{1-\gamma}}T(1-\gamma),$$

*where*

$$D(\gamma) = \frac{\log((\frac{\sigma}{\Delta_{max}^T})^2(1-\gamma)^2 \log \frac{1}{1-\gamma})}{\log \gamma}, C_1(\gamma) = e^{17} + 12 + 3\log\frac{1}{1-\gamma}, L_1(\gamma) = \frac{1152\log(\frac{1}{1-\gamma} + e^{17})\sigma^2}{\gamma^{1/(1-\gamma)}(\Delta_T(i))^2}.$$

**Corollary 4.2.** *When $\gamma$ is close to 1, $\gamma^{-\frac{1}{1-\gamma}}$ is around $e$. If the time horizon $T$ and number of breakpoints $B_T$ are known in advance, the discounted factor can be chosen as $\gamma = 1 - \frac{1}{\sigma}\sqrt{\frac{B_T}{T\log T}}$. If $B_T \ll T$,*

$$\frac{\sigma^2}{\Delta_{max}^T}(1-\gamma)^2 \log\frac{1}{1-\gamma} < \frac{\sigma/e}{\Delta_{max}^T}\sqrt{\frac{B_T}{T\log T}} < 1.$$

*We have*

$$\mathbb{E}[k_T(i)] = O(\sqrt{TB_T}(\log T)^{\frac{3}{2}}).$$

**Theorem 4.3** (SW-TS). *Let $\tau > 0$, for any suboptimal arm $i$,*

$$\mathbb{E}[k_T(i)] \leq B_T\tau + C_2(\tau)L_2(\tau)\frac{T}{\tau},$$

*where*

$$C_2(\tau) = e^{11} + 12 + 3\log\tau, L_2(\tau) = \frac{1152\log(\tau + e^{11})\sigma^2}{(\Delta_T(i))^2}.$$

**Corollary 4.4.** *If the time horizon $T$ and number of breakpoints $B_T$ are known in advance, the sliding window can be chosen as $\tau = \sigma\sqrt{T/B_T}\log T$, then*

$$\mathbb{E}[k_T(i)] = O(\sqrt{TB_T}\log T).$$

# 5 Proofs of Upper Bounds

Before giving the detailed proof, we discuss the main challenges in regret analysis of Thompson sampling in non-stationary setting. These challenges are addressed by Lemmas 5.1 to 5.3.

## 5.1 Challenges in Regret Analysis

Existing analyses of regret bounds for Thompson sampling (Agrawal & Goyal, 2013; Jin et al., 2021; 2022) decompose the regret into two parts. The first part of regret comes from the over-estimation of suboptimal arm, which can be dealt with by the concentration properties of the sampling distribution and rewards distribution. The second part is the under-estimation of the optimal arm, which mainly relies on bounding the following equation.

$$\sum_{t=1}^{T} \mathbb{E}[\frac{1 - p_{i,t}}{p_{i,t}} \mathbb{1}\{i_t = i_t^*, \theta_t(*) \leq \mu_t(*) - \epsilon_i\}], \tag{2}$$

where $p_{i,t} = \mathbb{P}(\theta_t(*) > \mu_t(*) - \epsilon_i)$ is the probability that the best arm will not be under-estimated from the mean reward by a margin $\epsilon_i$.

The first challenge is specific to the DS-TS algorithm. Unlike SW-TS, which completely forgets previous information after $\tau$ rounds following a breakpoint, DS-TS cannot **fully forget past information**. This makes it challenging to utilize the concentration properties of the reward distribution to bound regret comes from the over-estimate of the suboptimal arm. And this will further affect the analysis of Equation (2).

The second challenge is the **under-estimation of the optimal arm.** In stationary settings, $p_{i,t}$ changes only when the optimal arm is selected, Equation (2) can be bounded by the method proposed by Agrawal & Goyal (2013). However, the distribution of $\theta_t(*)$ may vary over time in non-stationary settings. It is challenging and nontrivial to obtain a tight bound of Equation (2).

To overcome the first challenges, we adjust the posterior variance to be $\frac{4\sigma^2}{N_t(\gamma,i)}$. This slightly larger variance is specifically designed for the $\sigma^2$-subGaussian distribution, which helps to bound $\mathbb{E}[\frac{1}{p_{i,t}}]$. Then, we define $U_t(\gamma, i)$, which serves a role similar to the upper confidence bound in the UCB algorithm. We solve this problem through Lemma 5.1 and Lemma 5.2.

For the second challenge, we use the new defined $U_t(\gamma, i)$ and employ a new regret decomposition for Equation (2) based on whether the event $\{N_t(\gamma, *) > L_1(\gamma)\}$ occurs. Intuitively, if $N_t(\gamma, *) > L_1(\gamma)$, $p_{i,t}$ is close to 1, which will lead to a sharp bound. If $N_t(\gamma, *) \leq L_1(\gamma)$, using Lemma A.3 we can also get the upper bound of Equation (2). We derive the upper bound of $\mathbb{E}[\frac{1}{p_{i,t}}]$ for non-stationary settings, with an extra logarithmic term compared with the stationary settings. The proof of Lemma 5.3 in Appendix B.3 demonstrates these details.

## 5.2 Proofs of Theorem 4.1

For arm $i \neq i_t^*$, we choose two threshold $x_t(i), y_t(i)$ such that $x_t(i) = \mu_t(i) + \frac{\Delta_t(i)}{3}, y_t(i) = \mu_t(*) - \frac{\Delta_t(i)}{3}$. Then $\mu_t(i) < x_t(i) < y_t(i) < \mu_t(*)$ and $y_t(i) - x_t(i) = \frac{\Delta_t(i)}{3}$. The history $\mathcal{F}_t$ is defined as the plays and rewards of the previous $t$ plays. $\hat{\mu}_t(\gamma, i), i_t$ and the distribution of $\theta_t(i)$ are determined by the history $\mathcal{F}_{t-1}$.

The abruptly changing setting is in fact piecewise-stationary. The rounds between two adjacent breakpoints is stable stationary. Based on this observation, we define the **pseudo-stationary phase** as

$$\mathcal{T}(\gamma) = \{t \leq T : \forall s \in (t - D(\gamma), t], \mu_s(\cdot) = \mu_t(\cdot)\}.$$

Let $\mathcal{S}(\gamma) = \{t \leq T : t \notin \mathcal{T}(\gamma)\}$. Note that, on the right side of any breakpoint, there will be at most $D(\gamma)$ rounds belonging to $\mathcal{S}(\gamma)$. Therefore, the number of elements in the set $\mathcal{S}(\gamma)$ has an upper bound $B_T D(\gamma)$, i.e.

$$|\mathcal{S}(\gamma)| \leq B_T D(\gamma) \tag{3}$$

Figure 1 shows $\mathcal{T}(\gamma)$ and $\mathcal{S}(\gamma)$ in two different situations.

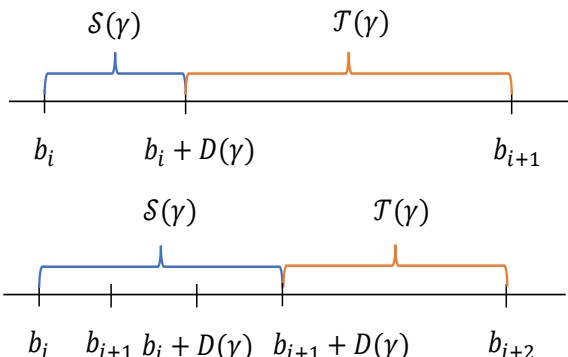

Figure 1: Illustration of $\mathcal{T}(\gamma)$ and $\mathcal{S}(\gamma)$ in two different situations. $b_i, b_{i+1}, b_{i+2}$ are the breakpoints. The situation that $b_{i+1} - b_i > D(\gamma)$ is shown in the top figure, and $b_{i+1} - b_i \leq D(\gamma)$ is in the bottom.

To facilitate the analysis, we define the following quantities

$$n = 6\sqrt{2} + 3\sqrt{1-\gamma}, A(\gamma) = \frac{n^2 \log(\frac{1}{1-\gamma})\sigma^2}{(\Delta_T(i))^2}, U_t(\gamma, i) = \sigma\sqrt{\frac{(1-\gamma)\log\frac{1}{1-\gamma}}{N_t(\gamma, i)}}. \tag{4}$$

Now we list some useful lemmas. The detailed proofs are provided in the appendix. The following lemma depicts that after finite rounds at the breakpoint, i.e., in the pseudo-stationary phase, the distance between $\mu_t(i)$ and discounted average of expectation for arm $i$ can be bounded by $U_t(\gamma, i)$. $U_t(\gamma, i)$ is analogous to the upper confidence bound in the UCB algorithm.

**Lemma 5.1.** Let $\ddot{\mu}_t(\gamma, i) = \frac{1}{N_t(\gamma, i)} \sum_{j=1}^{t} \gamma^{t-j} \mathbb{1}\{i_j = i\}\mu_j(i)$ denote the discounted average of expectation for arm $i$ at time step $t$. $\forall t \in \mathcal{T}(\gamma)$, the distance between $\mu_t(i)$ and $\ddot{\mu}_t(\gamma, i)$ is less than $U_t(\gamma, i)$.

$$|\mu_t(i) - \ddot{\mu}_t(\gamma, i)| \leq U_t(\gamma, i), \tag{5}$$

Using Lemma 5.1 and the self-normalized Hoeffding-type inequality for subGaussian distributions (Lemma A.1), we have the following lemma, which helps to bound regret comes from the over-estimation of suboptimal arm.

**Lemma 5.2.** $\forall t \in \mathcal{T}(\gamma), i \neq i_t^*$,

$$\mathbb{P}(\hat{\mu}_t(\gamma, i) > x_t(i), N_t(\gamma, i) > A(\gamma)) \leq (1-\gamma)^2$$

The following key lemma helps bound the regret comes from the under-estimation of the optimal arm. This is the most tricky part of analyzing TS. Note that, the proof in Trovo et al. (2020) does not prove the result of the following lemma.

**Lemma 5.3.** Let $p_{i,t} = \mathbb{P}(\theta_t(*) > y_t(i)|\mathcal{F}_{t-1})$. For any $t \in \mathcal{T}(\gamma)$ and $i \neq i_t^*$,

$$\sum_{t \in \mathcal{T}(\gamma)} \mathbb{E}[\frac{1-p_{i,t}}{p_{i,t}}\mathbb{1}\{i_t = i_t^*, \theta_t(i) < y_t(i)\}] \leq (e^{17} + 9 + 3\log\frac{1}{1-\gamma})T(1-\gamma)L_1(\gamma)\gamma^{-1/(1-\gamma)}.$$

Now we can give the detailed proof. The proof is in 5 steps:

**Step 1** We can divide the rounds $t \in \{1, ..., T\}$ into two parts: $\{t \in \mathcal{T}(\gamma)\}$ and $\{t \notin \mathcal{T}(\gamma)\}$. Equation (3) shows that the number of elements in the second part is smaller than $B_T D(\gamma)$, we have

$$\mathbb{E}[k_T(i)] \leq B_T D(\gamma) + \sum_{t \in \mathcal{T}(\gamma)} \mathbb{P}(i_t = i). \tag{6}$$

**Step 2** Then we consider the event $\{N_t(\gamma, i) > A(\gamma)\}$.

$$\sum_{t \in \mathcal{T}(\gamma)} \mathbb{P}(i_t = i) = \sum_{t \in \mathcal{T}(\gamma)} \mathbb{P}(i_t = i, N_t(\gamma, i) < A(\gamma)) + \sum_{t \in \mathcal{T}(\gamma)} \mathbb{P}(i_t = i, N_t(\gamma, i) > A(\gamma)).$$

We first bound $\sum_{t \in \mathcal{T}(\gamma)} \mathbb{P}(i_t = i, N_t(\gamma, i) < A(\gamma))$.

$$
\begin{aligned}
\sum_{t \in \mathcal{T}(\gamma)} \mathbb{P}(i_t = i, N_t(\gamma, i) < A(\gamma)) &= \sum_{t \in \mathcal{T}(\gamma)} \mathbb{E}\big[\mathbb{P}(i_t = i, N_t(\gamma, i) < A(\gamma) \mid \mathcal{F}_{t-1})\big] \\
&= \sum_{t \in \mathcal{T}(\gamma)} \mathbb{E}\big[\mathbb{E}\big[\mathbb{1}(i_t = i, N_t(\gamma, i) < A(\gamma) \mid \mathcal{F}_{t-1})\big]\big] \\
&= \sum_{t \in \mathcal{T}(\gamma)} \mathbb{E}\big[\mathbb{1}(i_t = i, N_t(\gamma, i) < A(\gamma))\big],
\end{aligned}
\tag{7}
$$

where the last equation uses the tower rule of expectation.

Using Lemma A.3, we have

$$\sum_{t \in \mathcal{T}(\gamma)} \mathbb{P}(i_t = i, N_t(\gamma, i) < A(\gamma)) \leq T(1 - \gamma)A(\gamma)\gamma^{-1/(1-\gamma)} \tag{8}$$

Therefore,

$$\mathbb{E}[k_T(i)] \leq T(1 - \gamma)A(\gamma)\gamma^{-1/(1-\gamma)} + B_T D(\gamma) + \sum_{t \in \mathcal{T}(\gamma)} \mathbb{P}(i_t = i, N_t(\gamma, i) > A(\gamma)) \tag{9}$$

**Step 3** Define $E_t(\gamma, i)$ as the event $\{i_t = i, N_t(\gamma, i) > A(\gamma)\}$. Define $E_t^\theta(i)$ as the event $\theta_t(i) < y_t(i)$. Equation (9) may be decomposed as follows:

$$
\begin{aligned}
\sum_{t \in \mathcal{T}(\gamma)} \mathbb{P}(E_t(\gamma, i)) = &\sum_{t \in \mathcal{T}(\gamma)} \mathbb{P}(E_t(\gamma, i), \hat{\mu}_t(\gamma, i) > x_t(i)) + \sum_{t \in \mathcal{T}(\gamma)} \mathbb{P}(E_t(\gamma, i), \hat{\mu}_t(\gamma, i) < x_t(i), \overline{E_t^\theta(i)}) \\
&+ \sum_{t \in \mathcal{T}(\gamma)} \mathbb{P}(E_t(\gamma, i), \hat{\mu}_t(\gamma, i) < x_t(i), E_t^\theta(i))
\end{aligned}
\tag{10}
$$

Using Lemma 5.2, the first part in Equation (10) can be bounded by $T(1 - \gamma)^2$.

**Step 4** Then we bound the second part in Equation (10). Use the fact that $N_t(\gamma, i)$ and $\hat{\mu}_t(i)$ are determined by the history $\mathcal{F}_{t-1}$, we have

$$
\begin{aligned}
&\sum_{t \in \mathcal{T}(\gamma)} \mathbb{P}(E_t(\gamma, i), \hat{\mu}_t(\gamma, i) < x_t(i), \overline{E_t^\theta(i)}) \\
&= \mathbb{E}\Bigg[\sum_{t \in \mathcal{T}(\gamma)} \mathbb{E}\big[\mathbb{1}\{i_t = i, N_t(\gamma, i) > A(\gamma), \hat{\mu}_t(\gamma, i) < x_t(i), \overline{E_t^\theta(i)}\} \mid \mathcal{F}_{t-1}\big]\Bigg] \\
&= \mathbb{E}\Bigg[\sum_{t \in \mathcal{T}(\gamma)} \mathbb{1}\{N_t(\gamma, i) > A(\gamma), \hat{\mu}_t(\gamma, i) < x_t(i)\}\mathbb{P}(i_t = i, \overline{E_t^\theta(i)} \mid \mathcal{F}_{t-1})\Bigg] \\
&\leq \mathbb{E}\Bigg[\sum_{t \in \mathcal{T}(\gamma)} \mathbb{1}\{N_t(\gamma, i) > A(\gamma), \hat{\mu}_t(\gamma, i) < x_t(i)\}\mathbb{P}(\theta_t(i) > y_t(i) \mid \mathcal{F}_{t-1})\Bigg].
\end{aligned}
\tag{11}
$$

Given the history $\mathcal{F}_{t-1}$ such that $N_t(\gamma, i) > A(\gamma)$ and $\hat{\mu}_t(\gamma, i) < x_t(i)$, we have

$$y_t(i) - \hat{\mu}_t(\gamma, i) > y_t(i) - x_t(i) = \frac{\Delta_t(i)}{3} \geq \frac{\Delta_T(i)}{3}.$$

Therefore,

$$\mathbb{P}(\theta_t(i) > y_t(i) \mid \mathcal{F}_{t-1})) \leq \mathbb{P}(\theta_t(i) - \hat{\mu}_t(\gamma, i) > \frac{\Delta_T(i)}{3}|\mathcal{F}_{t-1}) \leq \frac{1}{2}\exp(-\frac{(\Delta_T(i))^2 A(\gamma)}{72\sigma^2}) \leq \frac{1}{2}(1 - \gamma), \quad (12)$$

where the second inequality follows $\theta_t(i) \sim \mathcal{N}\big(\hat{\mu}_t(\gamma, i), \frac{4\sigma^2}{N_t(\gamma, i)}\big)$ and Fact 1.

For other $\mathcal{F}_{t-1}$, the indicator term $\mathbb{1}\{N_t(\gamma, i) > A(\gamma), \hat{\mu}_t(\gamma, i) < x_t(i)\}$ will be 0. Hence, we can bound the second part by $\frac{T}{2}(1 - \gamma)$

**Step 5** Finally, we focus the third term in Equation (10). Using Lemma A.2 and the fact that $p_{i,t}$ is fixed given $\mathcal{F}_{t-1}$,

$$\sum_{t \in \mathcal{T}(\gamma)} \mathbb{P}(E_t(\gamma, i), \hat{\mu}_t(\gamma, i) < x_t(i), E_t^\theta(i)) \leq \sum_{t \in \mathcal{T}(\gamma)} \mathbb{E}\left[\frac{1 - p_{i,t}}{p_{i,t}}\mathbb{P}(i_t = i_t^*, E_t^\theta(i)|\mathcal{F}_{t-1})\right]$$

$$= \sum_{t \in \mathcal{T}(\gamma)} \mathbb{E}\left[\mathbb{E}[\frac{1 - p_{i,t}}{p_{i,t}}\mathbb{1}\{i_t = i_t^*, E_t^\theta(i)|\mathcal{F}_{t-1}\}]\right]$$

$$= \sum_{t \in \mathcal{T}(\gamma)} \mathbb{E}[\frac{1 - p_{i,t}}{p_{i,t}}\mathbb{1}\{i_t = i_t^*, E_t^\theta(i)\}]$$

Then by Lemma 5.3, we have

$$\sum_{t \in \mathcal{T}(\gamma)} \mathbb{P}(E_t(\gamma, i), \hat{\mu}_t(\gamma, i) < x_t(i), E_t^\theta(i)) \leq (e^{17} + 9 + 3\log\frac{1}{1 - \gamma})T(1 - \gamma)L_1(\gamma)\gamma^{-1/(1-\gamma)}. \quad (13)$$

Substituting the results in Step 3-5 to Equation (10) and Equation (9),

$$\mathbb{E}[k_T(i)] \leq T(1 - \gamma)A(\gamma)\gamma^{-\frac{1}{1-\gamma}} + B_T D(\gamma) + 2T(1 - \gamma) + (e^{17} + 9 + 3\log\frac{1}{1 - \gamma})T(1 - \gamma)L_1(\gamma)\gamma^{-\frac{1}{1-\gamma}}$$

$$\leq B_T D(\gamma) + (e^{17} + 12 + 3\log\frac{1}{1 - \gamma})L_1(\gamma)\gamma^{-\frac{1}{1-\gamma}}T(1 - \gamma).$$

### 5.3 Proofs of Theorem 4.3

The proof of Theorem 4.3 is similar to Theorem 4.1. The main difference is that the pseudo-stationary phase is now defined as $\mathcal{T}(\tau) = \{t \leq T : \forall s \in (t - \tau, t], \mu_s(\cdot) = \mu_t(\cdot)\}$. Let

$$\ddot{\mu}_t(\tau, i) = \frac{1}{N_t(\tau, i)}\sum_{j=t-\tau+1}^{t}\mathbb{1}\{i_j = i\}\mu_j(i).$$

If $t \in \mathcal{T}(\tau)$,

$$\ddot{\mu}_t(\tau, i) = \frac{1}{N_t(\tau, i)}\sum_{j=t-\tau+1}^{t}\mathbb{1}\{i_j = i\}\mu_t(i) = \mu_t(i).$$

This means the bias $(U_t(\gamma, i))$ vanishes. We no longer need an $n$ related to $\tau$ to deal with the bias issue. We only need to define $A(\tau)$ as

$$A(\tau) = \frac{72\log(\tau)\sigma^2}{(\Delta_T(i))^2}.$$

We directly list the following two lemmas, corresponding to Lemma 5.2 and Lemma 5.3, respectively. The detailed proofs can be found in Appendix B.4 and Appendix B.5.

**Lemma 5.4.** $\forall t \in \mathcal{T}(\tau), t \neq i_t^*,$

$$\mathbb{P}(\hat{\mu}_t(\tau, i) > x_t(i), N_t(\tau, i) > A(\tau)) \leq \frac{1}{\tau^2}.$$

**Lemma 5.5.** *Let* $p_{i,t} = \mathbb{P}(\theta_t(*) > y_t(i)|\mathcal{F}_{t-1})$. *For any* $t \in \mathcal{T}(\tau)$ *and* $i \neq i_t^*,$

$$\sum_{t \in \mathcal{T}(\gamma)} \mathbb{E}[\frac{1 - p_{i,t}}{p_{i,t}} \mathbb{1}\{i_t = i_t^*, \theta_t(i) < y_t(i)\}] \leq (e^{11} + 9 + 3\log\tau)\frac{T}{\tau}L_2(\tau).$$

Let $\mathcal{S}(\tau) = \{t \leq T : t \notin \mathcal{T}(\tau)\}$. Similar to Equation (3), we have $|\mathcal{S}(\tau)| \leq B_T\tau$. Then the proof of step 1 is

$$\mathbb{E}[k_T(i)] \leq B_T\tau + \sum_{t \in \mathcal{T}(\tau)} \mathbb{P}(i_t = i).$$

The rest of the proof is nearly identical to the proof of Theorem 4.1.

## 6 Experiments

In this section, we empirically compare the performance of our method with state-of-the-art algorithms on Bernoulli and a Gaussian reward distributions. Specifically, we compare DS-TS and SW-TS with Thompson Sampling to evaluate the improvement obtained thanks to the employment of the discounted factor $\gamma$ and sliding window $\tau$. We also compare our method with the UCB method, DS-UCB and SW-UCB (Garivier & Moulines, 2011) to evaluate the effect of Thompson Sampling and UCB. Furthermore, we compare our method with some novel and efficient algorithms such as CUSUM (Liu et al., 2018), M-UCB (Cao et al., 2019) and SW-LB-SDA (Baudry et al., 2021). We measure the performance of each algorithm with the cumulative expected regret defined in Equation Equation (1). The expected regret is averaged over 100 independently runs. The 95% confidence interval is obtained by performing 100 independent runs and is depicted as a semi-transparent region in the figure.

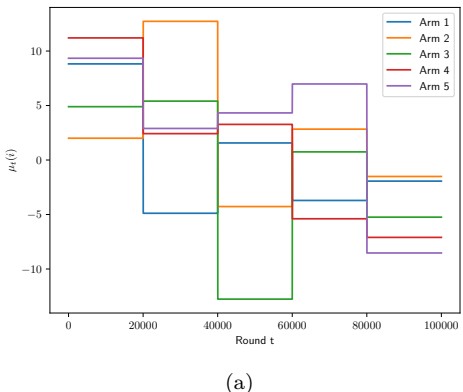

(a)

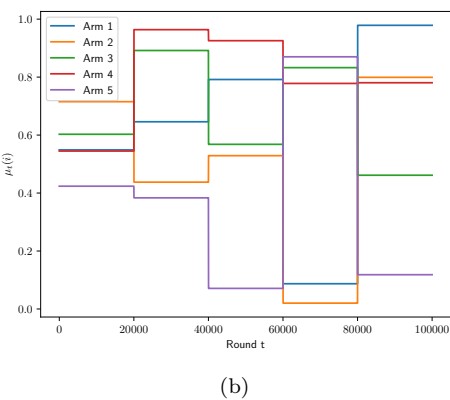

(b)

Figure 2: $K = 5, B_T = 5$. Gauss arms (a), Bernoulli arms (b).

### 6.1 Gaussian Arms

**Experimental setting for Gaussian arms** We fix the time horizon as $T = 100000$. The mean and variance are drawn from distributions $\mathcal{N}(0, 5^2)$ and $U(1, 5)$. For Gaussian rewards, we conduct two experiments. In the first experiment, we split the time horizon into 5 phases and use a number of arms $K = 5$. While in the second experiment, we split the time horizon into 10 phases and use a number of arms $K = 10$.

The analysis of SW-UCB and DS-UCB is conducted under the bounded reward assumption, but the algorithms can adapt to Gaussian scenarios. To achieve reasonable performance, it is necessary to adjust the discounted factor and the sliding-window appropriately. We use the settings recommended in (Baudry et al., 2021), where $\tau = 2(1 + 2\sigma)\sqrt{T \log(T)/B_T}$ for SW-UCB and $\gamma = 1 - 1/(4(1 + 2\sigma))\sqrt{B_T/T}$ for DS-UCB.

**Results** Figure 3 illustrates the performance of these algorithms for Gaussian rewards under two different settings. Notably, CUSUM and M-UCB are not applicable to Gaussian rewards: CUSUM is designed for Bernoulli distributions, while M-UCB assumes bounded distributions. The discounted methods tend to perform better than sliding-windows methods in Gaussian rewards. Among these algorithms, only our algorithms and SW-LB-SDA provide regret analysis for unbounded rewards. Our algorithm (DS-TS) and SW-LB-SDA have demonstrated highly competitive experimental performance.

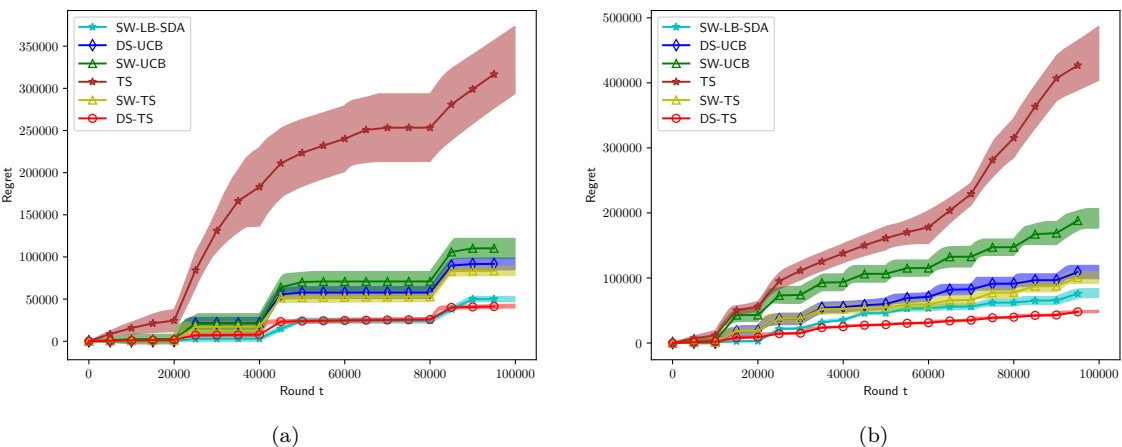

Figure 3: Gaussian arms. (a) $K = 5, B_T = 5$. (b) $K = 10, B_T = 10$

## 6.2 Bernoulli Arms

**Experimental setting for Bernoulli arms** The time horizon is set as $T = 100000$. We split the time horizon into $5, 10$ phases of equal length and use a number of arms $K = \{5, 10\}$, respectively.

For Bernoulli rewards, the expected value $\mu_t(i)$ of each arm $i$ is drawn from a uniform distribution over $[0, 1]$. In the stationary phase, the rewards distributions remain unchanged. The Bernoulli arms for each phase are generated as $\mu_t(i) \sim U(0, 1)$. Figure 2 depicts the expected rewards for Gaussian arms and Bernoulli arms with $K = 5$ and $B_T = 5$.

For Bernoulli distribution, we modify the Thompson sampling (step 3) in our algorithm as $\theta_t(i) \sim \mathcal{N}(\hat{\mu}_t(\gamma, i), \frac{1}{N_t(\gamma, i)})$ and $\theta_t(i) \sim \mathcal{N}(\hat{\mu}_t(\tau, i), \frac{1}{N_t(\tau, i)})$. Based on Corollary 4.2 and Corollary 4.4, we set $\gamma = 1 - \sqrt{\frac{B_T}{T \log T}}$ and $\tau = \sqrt{T/B_T} \log T$. To allow for fair comparison, DS-UCB uses the discount factor $\gamma = 1 - \sqrt{B_T/T}/4$, SW-UCB uses the sliding window $\tau = 2\sqrt{T \log T/B_T}$ suggested by (Garivier & Moulines, 2011). Based on (Baudry et al., 2021), we set $\tau = 2\sqrt{T \log(T)/B_T}$ for LB-SDA. For changepoint detection algorithm M-UCB, we set $w = 800, b = \sqrt{w/2 \log(2KT^2)}$ suggested by (Cao et al., 2019). But set the amount of exploration $\gamma = \sqrt{KB_T \log(T)/T}$. In practice, it has been found that using this value instead of the one guaranteed in (Cao et al., 2019) will improve empirical performance (Baudry et al., 2021). For CUSUM, following from (Liu et al., 2018), we set $\alpha = \sqrt{B_T/T \log(T/B_T)}$ and $h = \log(T/B_T)$. For our experiment settings, we choose $M = 50, \epsilon = 0.05$.

**Results** Figure 4 presents the results for Bernoulli arms in abruptly changing settings. It can be observed that our method (SW-TS) and SW-LB-SDA exhibit almost identical performance. Thompson Sampling, designed for stationary MAB problems, shows significant oscillations at the breakpoints. The changepoint

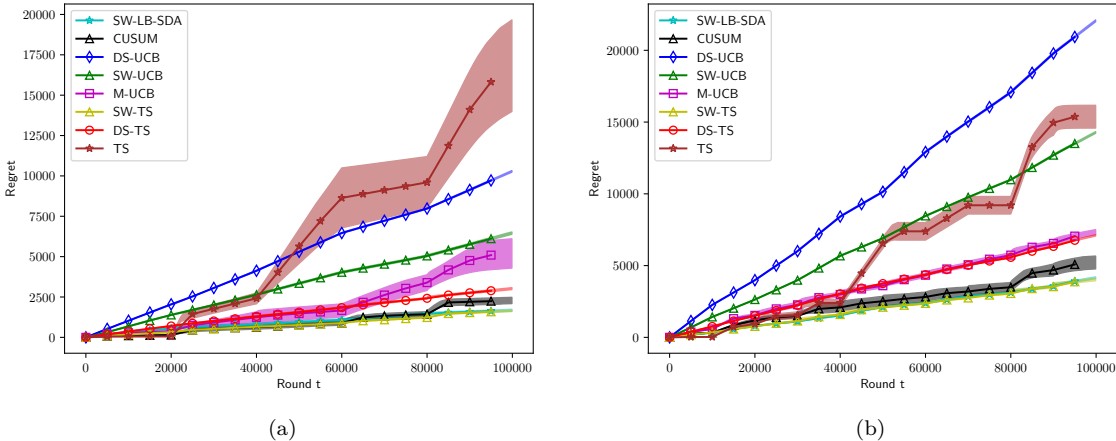

Figure 4: Bernoulli arms. Settings with $K = 5, B_T = 5$ (a), $K = 10, B_T = 10$ (b)

detection algorithm CUSUM (Liu et al., 2018) also shows competitive performance. Note that, our experiment does not satisfy the detectability assumption of CUSUM. As the number of arms and breakpoints increase, the performance of UCB-class algorithms (DS-UCB, SW-UCB) declines, while two TS-based algorithms (DS-TS, SW-TS) still work well.

**Storage and Compute Cost** These algorithms can be divided into three class: UCB, TS and SW-LB-SDA. At each round, UCB-class and TS-class algorithms require $O(K)$ storage and spend $O(K)$ time complexity for computational cost. However, for round $T$, SW-LB-SDA require $O(K(\log T)^2)$ storage and spend $O(K \log T)$ time cost. Although the experimental performance of SW-LB-SDA is similar to our algorithms, our algorithm has less storage space and lower computational complexity.

## 7 Conclusion

In this paper, we analyze the regret upper bound of the TS algorithm with Gaussian prior in non-stationary settings, filling a research gap in this field. Our approach builds upon previous works while tackling two key challenges specific to non-stationary environments: under-estimation of the optimal arm and the inability of DS-TS algorithm to fully forget previous information. Finally, we conduct some experiments to verify theory results. Below we discuss the results and propose directions for future research.

(1) The standard posterior update rule for Thompson Sampling has a sampling variance as $\frac{\sigma^2}{N}$. We use $\frac{4\sigma^2}{N}$ only for ease of analysis. While this discrepancy is significant only for relatively small values of $N$. It would be valuable to develop proof techniques that leverage the variance of standard Bayesian updates.

(2) Our regret upper bound includes an additional logarithmic term compared to DS-UCB and SW-UCB, along with coefficients of $e^{17}$ and $e^{11}$. It would be interesting to explore whether the additional logarithm and large coefficients are intrinsic to DS-TS and SW-TS algorithms or is a limitation of our analysis.

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

## A  Facts and Lemmas

In this section, we list some well-known lemmas.

Garivier & Moulines (2011) has derived a Hoeffding-type inequality for self-normalized means with a random number of summands. Their bound is for bounded distribution. Leveraging the properties of $\sigma$-subGaussian distributions, we have the following bound for $\sigma$-subGaussian. Recall that

$$N_t(\gamma, i) = \sum_{j=1}^t \gamma^{t-j} \mathbb{1}\{i_j = i\}, \hat{\mu}_t(\gamma, i) = \frac{1}{N_t(\gamma, i)} \sum_{j=1}^t \gamma^{t-j} X_j(i) \mathbb{1}\{i_j = i\}$$

$$\ddot{\mu}_t(\gamma, i) = \frac{1}{N_t(\gamma, i)} \sum_{j=1}^t \gamma^{t-j} \mathbb{1}\{i_j = i\} \mu_j(i)$$

**Lemma A.1.** *Let $t \in \mathcal{T}(\gamma), \delta > 0$,*

$$\mathbb{P}\Big(\frac{N_t(\gamma, i)(\hat{\mu}(\gamma, i) - \ddot{\mu}(\gamma, i))}{\sqrt{N_t(\gamma^2, i)}} > \delta\Big) \leq \log\big(\frac{1}{1-\gamma}\big) \exp\big(-\frac{3\delta^2}{8\sigma^2}\big).$$

*Let $t \in \mathcal{T}(\tau), \delta > 0$,*

$$\mathbb{P}\big(\sqrt{N_t(\tau, i)}(\hat{\mu}(\tau, i) - \mu_t(i)) > \delta\big) \leq \log \tau \exp\big(-\frac{3\delta^2}{8\sigma^2}\big),$$

The following inequality is the anti-concentration and concentration bound for Gaussian distributed random variables.

**Fact 1** (Abramowitz & Stegun (1964)). *For a Gaussian distributed random variable $X$ with mean $\mu$ and variance $\sigma^2$, for any $a > 0$*

$$\frac{1}{\sqrt{2\pi}} \frac{a}{1+a^2} e^{-a^2/2} \leq \mathbb{P}(X - \mu > a\sigma) \leq \frac{1}{a + \sqrt{a^2+4}} e^{-a^2/2}$$

*Since $\frac{1}{a+\sqrt{a^2+4}} \leq \frac{1}{2}$, we also have the following well-known result:*

$$\mathbb{P}(X - \mu > a\sigma) \leq \frac{1}{2} e^{-a^2/2}$$

The following lemma is adapted from Agrawal & Goyal (2013) and is often used in the analysis of Thompson Sampling, which can transform the probability of selecting the $i$th arm into the probability of selecting the optimal arm $i_t^*$.

**Lemma A.2.** *Let $p_{i,t} = \mathbb{P}(\theta_t(*) > y_t(i)|\mathcal{F}_{t-1})$. For any $A > 0$, $i \neq i_t^*$,*

$$\mathbb{P}(i_t = i, \theta_t(i) < y_t(i)|\mathcal{F}_{t-1}) \leq \frac{(1 - p_{i,t})}{p_{i,t}} \mathbb{P}(i_t = i_t^*, \theta_t(i) < y_t(i)|\mathcal{F}_{t-1})$$

**Lemma A.3** (Garivier & Moulines (2011)). *For any $i \in \{1, ..., K\}$, $\gamma \in (0, 1)$ and $A > 0$,*

$$\sum_{t=1}^{T} \mathbb{1}\{i_t = i, N_t(\gamma, i) < A\} \leq \lceil T(1-\gamma) \rceil A \gamma^{-1/(1-\gamma)},$$

$$\sum_{t=1}^{T} \mathbb{1}\{i_t = i, N_t(\tau, i) < A\} \leq \left\lceil \frac{T}{\tau} \right\rceil A.$$

# B  Detailed Proofs of Lemmas and Theorems

In this section, we provide the detailed proofs of Lemma 5.1, Lemma 5.2, Lemma 5.3, Lemma 5.4 and Lemma 5.5. The proof of Theorem 4.3 is almost identical to that of Theorem 4.1, so we have omitted the details of the proof.

## B.1  Proof of Lemma 5.1

Recall that $\ddot{\mu}_t(\gamma, i) = \frac{1}{N_t(\gamma, i)} \sum_{j=1}^{t} \gamma^{t-j} \mathbb{1}\{i_j = i\} \mu_j(i)$. Since $\ddot{\mu}_t(\gamma, i)$ is a convex combination of elements $\mu_j(i), j = 1, ..., t$, we have

$$|\mu_t(i) - \ddot{\mu}_t(\gamma, i)| \leq \Delta_{max}^T \tag{14}$$

We can write $\mu_t(i)$ as $\mu_t(i) = \frac{1}{N_t(\gamma, i)} \sum_{j=1}^{t} \gamma^{t-j} \mathbb{1}\{i_j = i\} \mu_t(i)$. Thus, we have

$$|\mu_t(i) - \ddot{\mu}_t(\gamma, i)| = \frac{1}{N_t(\gamma, i)} \left| \sum_{j=1}^{t} \gamma^{t-j} (\mu_j(i) - \mu_t(i)) \mathbb{1}\{i_j = i\} \right|.$$

Recall that $\mathcal{T}(\gamma) = \{t \leq T : \forall s \in (t - D(\gamma), t], \mu_s(\cdot) = \mu_t(\cdot)\}$. If $t \in \mathcal{T}(\gamma)$, we have $\mu_j(i) = \mu_t(i), \forall j \in (t - D(\gamma), t)$.

Therefore, $\forall t \in \mathcal{T}(\gamma)$, we have

$$\begin{aligned}
|\mu_t(i) - \ddot{\mu}_t(\gamma, i)| &= \frac{1}{N_t(\gamma, i)} \left| \sum_{j=1}^{t-D(\gamma)} \gamma^{t-j} (\mu_j(i) - \mu_t(i)) \mathbb{1}\{i_j = i\} \right| \\
&\leq \frac{\Delta_{max}^T}{N_t(\gamma, i)} \sum_{j=1}^{t-D(\gamma)} \gamma^{t-j} \mathbb{1}\{i_j = i\} \\
&= \frac{\Delta_{max}^T}{N_t(\gamma, i)} \gamma^{D(\gamma)} N_{t-D(\gamma)}(\gamma, i) \\
&\leq \frac{\Delta_{max}^T \gamma^{D(\gamma)}}{N_t(\gamma, i)(1-\gamma)}
\end{aligned}$$

where the last inequality follows from $N_{t-D(\gamma)}(\gamma, i) \leq \frac{1}{1-\gamma}$.

If $\frac{\gamma^{D(\gamma)}}{N_t(\gamma, i)(1-\gamma)} < 1$, $\frac{\gamma^{D(\gamma)}}{N_t(\gamma, i)(1-\gamma)} < \sqrt{\frac{\gamma^{D(\gamma)}}{N_t(\gamma, i)(1-\gamma)}}$, we have

$$|\mu_t(i) - \ddot{\mu}_t(\gamma, i)| \leq \Delta_{max}^T \sqrt{\frac{\gamma^{D(\gamma)}}{N_t(\gamma, i)(1-\gamma)}}.$$

If $\frac{\gamma^{D(\gamma)}}{N_t(\gamma, i)(1-\gamma)} \geq 1$, from Equation (14), we also have

$$|\mu_t(i) - \ddot{\mu}_t(\gamma, i)| \leq \Delta_{max}^T \leq \Delta_{max}^T \sqrt{\frac{\gamma^{D(\gamma)}}{N_t(\gamma, i)(1-\gamma)}}.$$

By the definition of $D(\gamma) = \frac{\log((\frac{\sigma}{\Delta_{max}^T})^2(1-\gamma)^2 \log\frac{1}{1-\gamma})}{\log\gamma}$,

$$|\mu_t(i) - \ddot{\mu}_t(\gamma, i)| \le \sigma\sqrt{\frac{(1-\gamma)\log\frac{1}{1-\gamma}}{N_t(\gamma, i)}}$$

## B.2 Proof of Lemma 5.2

From the definition of $n, A(\gamma), U_t(\gamma, i)$ in Equation (4), we can get

$$U_t(\gamma, i) = \frac{\sqrt{1-\gamma}\Delta_T(i)}{n}\sqrt{\frac{A(\gamma)}{N_t(\gamma, i)}}. \tag{15}$$

If $N_t(\gamma, i) > A(\gamma)$, $U_t(\gamma, i) < \frac{\sqrt{1-\gamma}}{n}\Delta_T(i)$. Thus, we have

$$\frac{\Delta_t(i)}{3} - U_t(\gamma, i) > \frac{\Delta_T(i)}{3} - \frac{\sqrt{1-\gamma}}{n}\Delta_T(i) = \frac{2\sqrt{2}}{n}\Delta_T(i). \tag{16}$$

Therefore,

$$\begin{aligned}
&\mathbb{P}(\hat{\mu}_t(\gamma, i) > \mu_t(i) + \frac{\Delta_t(i)}{3}, N_t(\gamma, i) > A(\gamma)) \\
&\stackrel{(a)}{\le} \mathbb{P}(\hat{\mu}_t(\gamma, i) - \ddot{\mu}_t(\gamma, i) > \frac{\Delta_t(i)}{3} - U_t(\gamma, i), N_t(\gamma, i) > A(\gamma)) \\
&\stackrel{(b)}{\le} \mathbb{P}(\hat{\mu}_t(\gamma, i) - \ddot{\mu}_t(\gamma, i) > \frac{2\sqrt{2}}{n}\Delta_T(i), N_t(\gamma, i) > A(\gamma)) \\
&\stackrel{(c)}{\le} \mathbb{P}(\frac{N_t(\gamma, i)(\hat{\mu}_t(\gamma, i) - \ddot{\mu}_t(\gamma, i))}{\sqrt{N_t(\gamma^2, i)}} > \frac{2\sqrt{2}}{n}\Delta_T(i)\sqrt{A(\gamma)}) \\
&\stackrel{(d)}{\le} \log\frac{1}{1-\gamma}\exp(-\frac{3(\Delta_T(i))^2}{n^2\sigma^2}A(\gamma)) \\
&\le (1-\gamma)^3\log\frac{1}{1-\gamma}
\end{aligned} \tag{17}$$

where (a) uses Lemma 5.1, (b) uses Equation (16), (c) follows from $N_t(\gamma, i) > N_t(\gamma^2, i)$, (d) uses Lemma A.1. Since $(1-\gamma)\log\frac{1}{1-\gamma} \le \frac{1}{e} < 1$, this ends the proof.

## B.3 Proof of Lemma 5.3

This proof is adapted from Agrawal & Goyal (2013) for the stationary settings. However, there are some technical problems that are difficult to overcome in non-stationary settings. The tricky problem is to lower bound the probability of the mean's estimation of optimal arm Equation (21). By designing the function $U_t(\gamma, i)$ and decomposing the regret to use Lemma A.3 again, we solve this challenge. We use blue font to emphasize the techniques used in the proof.

The proof is in 3 steps.

**Step 1** We first prove that $\mathbb{E}[\frac{1}{p_{i,t}}]$ has an upper bound independent of $t$.

Define a Bernoulli experiment as sampling from $\mathcal{N}(\hat{\mu}_t(*), \frac{4\sigma^2}{N_t(\gamma, *)})$, where success implies that $\theta_t(*) > y_t(i)$. Let $G_t$ denote the number of experiments performed when the event $\{\theta_t(*) > y_t(i)\}$ first occurs. Then

$$\mathbb{E}[\frac{1}{p_{i,t}}] = \mathbb{E}[\mathbb{E}[G_t|\mathcal{F}_{t-1}]] = \mathbb{E}[G_t]$$

Let $z = \sqrt{\log r} + \frac{1}{2}$ ($r \geq 1$ is an integer ) and let $\mathrm{MAX}_r$ denote the maximum of $r$ independent Bernoulli experiment. Then

$$
\begin{aligned}
\mathbb{P}(G_t \leq r) &\geq \mathbb{P}(\mathrm{MAX}_r > \hat{\mu}_t(*) + \frac{z \cdot 2\sigma}{\sqrt{N_t(\gamma, *)}} \geq y_t(i)) \\
&= \mathbb{E}[\mathbb{E}[\mathbb{1}\{\mathrm{MAX}_r > \hat{\mu}_t(*) + \frac{z \cdot 2\sigma}{\sqrt{N_t(\gamma, *)}} \geq y_t(i)\}|\mathcal{F}_{t-1}]] \\
&= \mathbb{E}[\mathbb{1}\{\hat{\mu}_t(*) + \frac{z \cdot 2\sigma}{\sqrt{N_t(\gamma, *)}} \geq y_t(i)\}\mathbb{P}(\mathrm{MAX}_r > \hat{\mu}_t(*) + \frac{z \cdot 2\sigma}{\sqrt{N_t(\gamma, *)}}|\mathcal{F}_{t-1})]
\end{aligned}
\tag{18}
$$

Using Fact 1,

$$
\begin{aligned}
\mathbb{P}(\mathrm{MAX}_r > \hat{\mu}_t(*) + \frac{z \cdot 2\sigma}{\sqrt{N_t(\gamma, *)}}|\mathcal{F}_{t-1}) &\geq 1 - (1 - \frac{1}{\sqrt{2\pi}} \frac{z}{z^2 + 1} e^{-z^2/2})^r \\
&= 1 - (1 - \frac{1}{\sqrt{2\pi}} \frac{\sqrt{\log r} + \frac{1}{2}}{(\sqrt{\log r} + \frac{1}{2})^2 + 1} \frac{e^{-1/4 - \sqrt{\log r}/2}}{\sqrt{r}})^r \\
&\geq 1 - e^{-\frac{\sqrt{r} e^{-\sqrt{\log r}/2}}{e^{0.25}\sqrt{2\pi}(\sqrt{\log r} + 1)}}
\end{aligned}
\tag{19}
$$

For any $r \geq e^{17}$, $e^{-\frac{\sqrt{r} e^{-\sqrt{\log r}/2}}{e^{0.25}\sqrt{2\pi}(\sqrt{\log r} + 1)}} \leq \frac{1}{r^2}$. Hence, for any $r \geq e^{17}$,

$$
\mathbb{P}(\mathrm{MAX}_r > \hat{\mu}_t(*) + \frac{z \cdot 2\sigma}{\sqrt{N_t(\gamma, *)}}|\mathcal{F}_{t-1}) \geq 1 - \frac{1}{r^2}.
$$

Therefore, for any $r \geq e^{17}$,

$$
\mathbb{P}(G_t \leq r) \geq (1 - \frac{1}{r^2})\mathbb{P}(\hat{\mu}_t(*) + \frac{z}{\sqrt{N_t(\gamma, *)}} \geq y_t(i))
$$

Next, we apply Lemma A.1 to lower bound $\mathbb{P}(\hat{\mu}_t(*) + \frac{z \cdot 2\sigma}{\sqrt{N_t(\gamma, *)}} \geq y_t(i))$.

$$
\begin{aligned}
\mathbb{P}(\hat{\mu}_t(*) + \frac{z \cdot 2\sigma}{\sqrt{N_t(\gamma, *)}} \geq y_t(i)) &\geq 1 - \mathbb{P}(\hat{\mu}_t(*) + \frac{z \cdot 2\sigma}{\sqrt{N_t(\gamma, *)}} \leq \mu_t(*)) \\
&\geq 1 - \mathbb{P}(\hat{\mu}_t(*) - \ddot{\mu}_t(*) \leq U_t(\gamma, *) - \frac{z \cdot 2\sigma}{\sqrt{N_t(\gamma, *)}})
\end{aligned}
\tag{20}
$$

Since $U_t(\gamma, *) = \frac{\sigma\sqrt{(1-\gamma)\log\frac{1}{1-\gamma}}}{\sqrt{N_t(\gamma, *)}}$, $z = \log r + \frac{1}{2}$,

$$
U_t(\gamma, *) - \frac{z \cdot 2\sigma}{\sqrt{N_t(\gamma, *)}} = \frac{\sigma\sqrt{(1-\gamma)\log\frac{1}{1-\gamma}} - \sigma - 2\sigma\sqrt{\log r}}{\sqrt{N_t(\gamma, *)}} < -\frac{2\sigma\sqrt{\log r}}{\sqrt{N_t(\gamma, *)}}.
$$

Then we have

$$
\begin{aligned}
\mathbb{P}(\hat{\mu}_t(*) + \frac{z \cdot 2\sigma}{\sqrt{N_t(\gamma, *)}} \geq y_t(i)) &\geq 1 - \mathbb{P}(\hat{\mu}_t(*) - \ddot{\mu}_t(*) < -\frac{2\sigma\sqrt{\log r}}{\sqrt{N_t(\gamma, *)}}) \\
&\geq 1 - \log(\frac{1}{1-\gamma})e^{-\frac{3}{2}\log r} \\
&\geq 1 - \log\frac{1}{1-\gamma}\frac{1}{r^{1.5}}.
\end{aligned}
\tag{21}
$$

Substituting, for any $r > e^{17}$,

$$
\mathbb{P}(G_t \leq r) \geq 1 - \log\frac{1}{1-\gamma}\frac{1}{r^{1.5}} - \frac{1}{r^2}
\tag{22}
$$

Therefore,

$$\mathbb{E}[G_t] = \sum_{r=0}^{\infty} \mathbb{P}(G_t \geq r)$$

$$\leq 1 + e^{17} + \sum_{r > e^{17}} (\log \frac{1}{1-\gamma} \frac{1}{r^{1.5}} + \frac{1}{r^2})$$

$$\leq e^{17} + 3 + 3\log \frac{1}{1-\gamma}$$

This proves a bound of $\mathbb{E}[\frac{1}{p_{i,t}}] \leq e^{17} + 3 + 3\log \frac{1}{1-\gamma}$ independent of $t$.

**Step 2**. Define $L(\gamma) = \frac{1152\log(\frac{1}{1-\gamma}+e^{17})\sigma^2}{(\Delta_T(i))^2}$. We consider the upper bound of $\mathbb{E}[\frac{1}{p_{i,t}}]$ when $N_t(\gamma, *) > L(\gamma)$.

$$\mathbb{P}(G_t \leq r) \geq \mathbb{P}(\text{MAX}_r > \hat{\mu}_t(*) + \frac{z \cdot 2\sigma}{\sqrt{N_t(\gamma, *)}} - \frac{\Delta_t(i)}{6} \geq y_t(i))$$

$$= \mathbb{E}[\mathbb{1}\{\hat{\mu}_t(*) + \frac{z \cdot 2\sigma}{\sqrt{N_t(\gamma, *)}} - \frac{\Delta_t(i)}{6} \geq y_t(i)\}\mathbb{P}(\text{MAX}_r > \hat{\mu}_t(*) + \frac{z \cdot 2\sigma}{\sqrt{N_t(\gamma, *)}} - \frac{\Delta_t(i)}{6}|\mathcal{F}_{t-1})] \tag{23}$$

Now, since $N_t(\gamma, *) > L(\gamma), \frac{1}{\sqrt{N_t(\gamma,*)}} < \frac{\Delta_t(i)}{48\sqrt{\log(\frac{1}{1-\gamma}+e^{17})}\sigma}$. Therefore, for any $r \leq (\frac{1}{1-\gamma} + e^{17})^2$,

$$\frac{z \cdot 2\sigma}{\sqrt{N_t(\gamma, *)}} - \frac{\Delta_t(i)}{6} = \frac{2\sigma\sqrt{\log r} + \sigma}{\sqrt{N_t(\gamma, *)}} - \frac{\Delta_t(i)}{6} \leq -\frac{\Delta_t(i)}{12}.$$

Using Fact 1,

$$\mathbb{P}(\theta_t(i) > \hat{\mu}_t(i) - \frac{\Delta_t(i)}{12}|\mathcal{F}_{t-1}) \leq 1 - \frac{1}{2}e^{-\frac{N_t(\gamma,*)}{4\sigma^2}\frac{\Delta_t(i)^2}{288}} \geq 1 - \frac{1}{2(1/(1-\gamma)+e^{17})}.$$

This implies

$$\mathbb{P}(\text{MAX}_r > \hat{\mu}_t(*) + \frac{z}{\sqrt{N_t(\gamma, *)}} - \frac{\Delta_t(i)}{6}|\mathcal{F}_{t-1}) \geq 1 - \frac{1}{2^r(1/(1-\gamma)+e^{17})^r}.$$

Also, apply the self-normalized Hoeffding-type inequality,

$$\mathbb{P}(\hat{\mu}_t(*) + \frac{z \cdot 2\sigma}{\sqrt{N_t(\gamma, *)}} - \frac{\Delta_t(i)}{6} \geq y_t(i)) \geq 1 - \mathbb{P}(\hat{\mu}_t(*) \leq \mu_t(*) - \frac{\Delta_t(i)}{6})$$

$$\geq 1 - \mathbb{P}(\hat{\mu}_t(*) - \ddot{\mu}_t(*) \geq -U_t(\gamma, *) + \frac{\Delta_t(i)}{6})$$

$$> 1 - \mathbb{P}(\hat{\mu}_t(*) - \ddot{\mu}_t(*) \geq \frac{\Delta_T(i)}{8}\sqrt{\frac{L(\gamma)}{N_t(\gamma, *)}})$$

$$\geq 1 - \log(\frac{1}{1-\gamma} + e^{17})\frac{1}{(1/(1-\gamma)+e^{17})^3}.$$

Let $\gamma' = (\frac{1}{1-\gamma} + e^{17})^2$. Therefore, for any $1 \leq r \leq \gamma'$,

$$\mathbb{P}(G_t \leq r) \geq 1 - \frac{1}{2^r \gamma'^{r/2}} - \log(\frac{1}{1-\gamma} + e^{17})\frac{1}{\gamma'^{1.5}}.$$

When $r \geq \gamma' > e^{17}$, we can use Equation (22) to obtain,

$$\mathbb{P}(G_t \leq r) \geq 1 - \log\frac{1}{1-\gamma}\frac{1}{r^{1.5}} - \frac{1}{r^2}$$

Combining these results,

$$\mathbb{E}[G_t] \le \sum_{r=0}^{\infty} \mathbb{P}(G_t \ge r)$$

$$\le 1 + \sum_{r=1}^{\gamma'} \mathbb{P}(G_t \ge r) + \sum_{r=\gamma'}^{\infty} \mathbb{P}(G_t \ge r)$$

$$\le 1 + \sum_{r=1}^{\gamma'} \left(\frac{1}{2^r \gamma'^{r/2}} + \log(\frac{1}{1-\gamma} + e^{17}) \frac{1}{\gamma'^{1.5}}\right) + \sum_{r=\gamma'}^{\infty} \left(\log \frac{1}{1-\gamma} \frac{1}{r^{1.5}} + \frac{1}{r^2}\right)$$

$$\le 1 + \frac{1}{2\sqrt{\gamma'}} + \log(\frac{1}{1-\gamma} + e^{17}) \frac{1}{\sqrt{\gamma'}} + \frac{2}{\gamma'} + \log(\frac{1}{1-\gamma}) \frac{3}{\sqrt{\gamma'}}$$

$$\le 1 + 6(1-\gamma) \log(\frac{1}{1-\gamma} + e^{17}).$$

Therefore, when $N_t(\gamma, *) > L(\gamma)$, it holds that

$$\mathbb{E}[\frac{1}{p_{i,t}}] - 1 = \mathbb{E}[G_t] - 1 \le 6(1-\gamma) \log(\frac{1}{1-\gamma} + e^{17}).$$

**Step 3**  Let $\mathcal{A}(\gamma, *) = \{t \in \{1, ..., T\} : N_t(\gamma, *) \le L(\gamma)\}$.

$$\sum_{t \in \mathcal{T}(\gamma)} \mathbb{E}[\frac{1 - p_{i,t}}{p_{i,t}} \mathbb{1}\{i_t = i_t^*, \theta_t(i) < y_t(i)\}]$$

$$\le \left(\sum_{t \in \mathcal{T}(\gamma) \cap \mathcal{A}(\gamma, *)} + \sum_{t \in \mathcal{T}(\gamma) \setminus \mathcal{A}(\gamma, *)}\right) \mathbb{E}\left[\frac{1 - p_{i,t}}{p_{i,t}} \mathbb{1}\{i_t = i_t^*, \theta_t(i) < y_t(i)\}\right]$$

$$\le \left|\{t : i_t = i_t^*, N_t(\gamma, *) \le L(\gamma)\}\right|(e^{17} + 3 + 3\log\frac{1}{1-\gamma}) + \sum_{t \in \mathcal{T}(\gamma) \setminus \mathcal{A}(\gamma, *)} \mathbb{E}\left[\frac{1 - p_{i,t}}{p_{i,t}}\right] \qquad (24)$$

$$\le T(1-\gamma)L(\gamma)\gamma^{-1/(1-\gamma)}(e^{17} + 3 + 3\log\frac{1}{1-\gamma}) + 6T(1-\gamma)\log(\frac{1}{1-\gamma} + e^{17})$$

$$\le (e^{17} + 9 + 3\log\frac{1}{1-\gamma})T(1-\gamma)L(\gamma)\gamma^{-1/(1-\gamma)}.$$

Lemma A.1 has a stricter upper bound as $\frac{\log\frac{1}{1-\gamma}}{\log(1+\eta)} \exp(-\frac{1}{2\sigma^2}(1 - \frac{\eta^2}{16}))$. Suppose the variance of Thompson sampling is $\frac{\xi\sigma^2}{N_t(\gamma,i)}$. The lower bound of Equation (21) becomes

$$\frac{\log\frac{1}{1-\gamma}}{\log(1+\eta)} \exp(-\frac{\xi\log r}{2}(1 - \frac{\eta^2}{16})).$$

To ensure that $\mathbb{E}[G_t]$ has a finite upper bound, our analysis method requires $\xi > 2$, i.e. the sampling variance needs to be strictly greater than $\frac{2\sigma^2}{N_t(\gamma,i)}$.

## B.4 Proof of Lemma 5.4

Recall that $A(\tau) = \frac{72 \log(\tau) \sigma^2}{(\Delta_T(i))^2}$. Using Lemma A.1, we have

$$
\begin{aligned}
\mathbb{P}(\hat{\mu}_t(\tau, i) > x_t(i), N_t(\tau, i) > A(\tau)) &= \mathbb{P}(\hat{\mu}_t(\tau, i) - \mu_t(i) > \frac{\Delta_t(i)}{3}, N_t(\tau, i) > A(\gamma)) \\
&\leq \mathbb{P}(\sqrt{N_t(\tau, i)}(\hat{\mu}_t(\tau, i) - \mu_t(i)) > \frac{\Delta_T(i)}{3}\sqrt{A(\gamma)}) \\
&\leq \log \tau \exp(-\frac{3(\Delta_T(i))^2}{72\sigma^2} A(\gamma)) \\
&\leq \frac{1}{\tau^2}
\end{aligned}
\tag{25}
$$

## B.5 Proof of Lemma 5.5

The proof is similar to the proof of Lemma 5.3.

**Step 1** We first prove that $\mathbb{E}[\frac{1}{p_{i,t}}]$ has an upper bound independent of $t$.

Let $z = \sqrt{\log r}$ ($r \geq 1$ is an integer ). Then

$$
\begin{aligned}
\mathbb{P}(G_t \leq r) &\geq \mathbb{P}(\mathrm{MAX}_r > \hat{\mu}_t(*) + \frac{z \cdot 2\sigma}{\sqrt{N_t(\tau, *)}} \geq y_t(i)) \\
&= \mathbb{E}[\mathbb{1}\{\hat{\mu}_t(*) + \frac{z \cdot 2\sigma}{\sqrt{N_t(\tau, *)}} \geq y_t(i)\}\mathbb{P}(\mathrm{MAX}_r > \hat{\mu}_t(*) + \frac{z \cdot 2\sigma}{\sqrt{N_t(\tau, *)}}|\mathcal{F}_{t-1})]
\end{aligned}
\tag{26}
$$

Using Fact 1,

$$
\begin{aligned}
\mathbb{P}(\mathrm{MAX}_r > \hat{\mu}_t(*) + \frac{z \cdot 2\sigma}{\sqrt{N_t(\tau, *)}}|\mathcal{F}_{t-1}) &\geq 1 - (1 - \frac{1}{\sqrt{2\pi}}\frac{z}{z^2 + 1}e^{-z^2/2})^r \\
&= 1 - (1 - \frac{1}{\sqrt{2\pi}}\frac{\sqrt{\log r}}{(\sqrt{\log r})^2 + 1}\frac{1}{\sqrt{r}})^r \\
&\geq 1 - e^{-\frac{\sqrt{r}}{\sqrt{2\pi}(\sqrt{\log r}+1)}}
\end{aligned}
\tag{27}
$$

For any $r \geq e^{11}$, $e^{-\frac{\sqrt{r}}{\sqrt{2\pi}(\sqrt{\log r}+1)}} \leq \frac{1}{r^2}$. Hence, for any $r \geq e^{11}$,

$$
\mathbb{P}(\mathrm{MAX}_r > \hat{\mu}_t(*) + \frac{z \cdot 2\sigma}{\sqrt{N_t(\tau, *)}}|\mathcal{F}_{t-1}) \geq 1 - \frac{1}{r^2}.
$$

Therefore, for any $r \geq e^{11}$,

$$
\mathbb{P}(G_t \leq r) \geq (1 - \frac{1}{r^2})\mathbb{P}(\hat{\mu}_t(*) + \frac{z}{\sqrt{N_t(\tau, *)}} \geq y_t(i))
$$

Next, we apply Lemma A.1 to lower bound $\mathbb{P}(\hat{\mu}_t(*) + \frac{z \cdot 2\sigma}{\sqrt{N_t(\tau, *)}} \geq y_t(i))$.

$$
\begin{aligned}
\mathbb{P}(\hat{\mu}_t(*) + \frac{z \cdot 2\sigma}{\sqrt{N_t(\tau, *)}} \geq y_t(i)) &\geq 1 - \mathbb{P}(\hat{\mu}_t(*) + \frac{z \cdot 2\sigma}{\sqrt{N_t(\tau, *)}} \leq \mu_t(*)) \\
&\geq 1 - \mathbb{P}(\hat{\mu}_t(*) - \mu_t(*) < -\frac{2\sigma\sqrt{\log r}}{\sqrt{N_t(\tau, *)}}) \\
&\geq 1 - \log \tau e^{-\frac{3}{2}\log r} \\
&= 1 - \log \tau \frac{1}{r^{1.5}}.
\end{aligned}
$$

Substituting, for any $r > e^{11}$,

$$\mathbb{P}(G_t \leq r) \geq 1 - \log \tau \frac{1}{r^{1.5}} - \frac{1}{r^2} \tag{28}$$

Therefore,

$$\mathbb{E}[G_t] = \sum_{r=0}^{\infty} \mathbb{P}(G_t \geq r)$$

$$\leq 1 + e^{11} + \sum_{r > e^{11}} (\log \tau \frac{1}{r^{1.5}} + \frac{1}{r^2})$$

$$\leq e^{11} + 3 + 3 \log \tau$$

This proves a bound of $\mathbb{E}[\frac{1}{p_{i,t}}] \leq e^{11} + 3 + 3 \log \tau$ independent of $t$.

**Step 2**. Define $L(\tau) = \frac{1152 \log(\tau + e^{11}) \sigma^2}{(\Delta_T(i))^2}$. We consider the upper bound of $\mathbb{E}[\frac{1}{p_{i,t}}]$ when $N_t(\tau, *) > L(\tau)$.

$$\mathbb{P}(G_t \leq r) \geq \mathbb{P}(\text{MAX}_r > \hat{\mu}_t(*) + \frac{z \cdot 2\sigma}{\sqrt{N_t(\tau, *)}} - \frac{\Delta_t(i)}{6} \geq y_t(i))$$

$$= \mathbb{E}[\mathbb{1}\{\hat{\mu}_t(*) + \frac{z \cdot 2\sigma}{\sqrt{N_t(\tau, *)}} - \frac{\Delta_t(i)}{6} \geq y_t(i)\} \mathbb{P}(\text{MAX}_r > \hat{\mu}_t(*) + \frac{z \cdot 2\sigma}{\sqrt{N_t(\tau, *)}} - \frac{\Delta_t(i)}{6} | \mathcal{F}_{t-1})] \tag{29}$$

Now, since $N_t(\tau, *) > L(\tau), \frac{1}{\sqrt{N_t(\tau, *)}} < \frac{\Delta_t(i)}{48\sqrt{\log(\tau + e^{11})}\sigma}$. Therefore, for any $r \leq (\tau + e^{11})^2$,

$$\frac{z \cdot 2\sigma}{\sqrt{N_t(\tau, *)}} - \frac{\Delta_t(i)}{6} = \frac{2\sigma\sqrt{\log r} + \sigma}{\sqrt{N_t(\tau, *)}} - \frac{\Delta_t(i)}{6} \leq -\frac{\Delta_t(i)}{12}.$$

Using Fact 1,

$$\mathbb{P}(\theta_t(i) > \hat{\mu}_t(i) - \frac{\Delta_t(i)}{12} | \mathcal{F}_{t-1}) \leq 1 - \frac{1}{2} e^{-\frac{N_t(\tau, *)}{4\sigma^2} \frac{\Delta_t(i)^2}{288}} \geq 1 - \frac{1}{2(\tau + e^{11})}.$$

This implies

$$\mathbb{P}(\text{MAX}_r > \hat{\mu}_t(*) + \frac{z}{\sqrt{N_t(\tau, *)}} - \frac{\Delta_t(i)}{6} | \mathcal{F}_{t-1}) \geq 1 - \frac{1}{2^r (\tau + e^{11})^r}.$$

Also, apply Lemma A.1,

$$\mathbb{P}(\hat{\mu}_t(*) + \frac{z}{\sqrt{N_t(\tau, *)}} - \frac{\Delta_t(i)}{6} \geq y_t(i)) \geq 1 - \mathbb{P}(\hat{\mu}_t(*) - \mu_t(*) \geq \frac{\Delta_t(i)}{6})$$

$$\geq 1 - \log(\tau + e^{11}) \frac{1}{(\tau + e^{11})^3}.$$

Let $\tau' = (\tau + e^{11})^2$. Therefore, for any $1 \leq r \leq \tau'$,

$$\mathbb{P}(G_t \leq r) \geq 1 - \frac{1}{2^r \tau'^{r/2}} - \log(\tau + e^{11}) \frac{1}{\tau'^{1.5}}.$$

When $r \geq \tau' > e^{11}$, we can use Equation (22) to obtain,

$$\mathbb{P}(G_t \leq r) \geq 1 - \log \tau \frac{1}{r^{1.5}} - \frac{1}{r^2}$$

Combining these results,

$$\mathbb{E}[G_t] \leq \sum_{r=0}^{\infty} \mathbb{P}(G_t \geq r)$$

$$\leq 1 + \sum_{r=1}^{\tau'} \mathbb{P}(G_t \geq r) + \sum_{r=\tau'}^{\infty} \mathbb{P}(G_t \geq r)$$

$$\leq 1 + \frac{6}{\tau} \log(\tau + e^{11}).$$

Therefore, when $N_t(\tau, *) > L(\tau)$, it holds that

$$\mathbb{E}[\frac{1}{p_{i,t}}] - 1 = \mathbb{E}[G_t] - 1 \leq \frac{6}{\tau}\log(\tau + e^{11}).$$

**Step 3**  Let $\mathcal{A}(\tau, *) = \{t \in \{1, ..., T\} : N_t(\tau, *) \leq L(\tau)\}$ and $C = e^{11} + 9$.

$$\sum_{t \in \mathcal{T}(\tau)} \mathbb{E}[\frac{1 - p_{i,t}}{p_{i,t}}\mathbb{1}\{i_t = i_t^*, \theta_t(i) < y_t(i)\}]$$

$$\leq \sum_{t \in \mathcal{T}(\tau) \cap \mathcal{A}(\tau, *)} \mathbb{E}[\frac{1 - p_{i,t}}{p_{i,t}}\mathbb{1}\{i_t = i_t^*, \theta_t(i) < y_t(i)\}] + \sum_{t \in \mathcal{T}(\tau) \setminus \mathcal{A}(\tau, *)} \mathbb{E}[\frac{1 - p_{i,t}}{p_{i,t}}\mathbb{1}\{i_t = i_t^*, \theta_t(i) < y_t(i)\}]$$

$$\leq |\{t \in \{1, ..., T\} : i_t = i_t^*, N_t(\tau, *) \leq L(\tau)\}|(e^{11} + 3 + 3\log\tau) + \sum_{t \in \mathcal{T}(\tau) \setminus \mathcal{A}(\tau, *)} \mathbb{E}[\frac{1 - p_{i,t}}{p_{i,t}}] \tag{30}$$

$$\leq \frac{T}{\tau}L(\tau)(e^{11} + 3 + 3\log\tau) + 6\frac{T}{\tau}\log(\tau + e^{11})$$

$$\leq (e^{11} + 9 + 3\log\tau)\frac{T}{\tau}L(\tau).$$

## C   Incorrect Proof of SW-TS with Beta Priors

Here, we discuss the mistakes in proof of Trovo et al. (2020). It is precisely because of these errors that they bypassed the analysis of under-estimation of the optimal arm ( i.e. Lemma 5.3).

We first define the same notions in Trovo et al. (2020).

Let $\mathcal{F}_\phi' := \{t : b_{\phi-1} + \tau \leq t < b_\phi\}$, $b_\phi$ is the $\phi$-th breakpoints. $T_i(\mathcal{F}_\phi') := \sum_{t \in \mathcal{F}_\phi'} \mathbb{1}\{i_t = i, i \neq i_\phi^*\}$ denote the number of times a suboptimal arm is played during phase $\mathcal{F}_\phi'$.

$T_{i,t,r} := \sum_{s=\max\{t-\tau+1, 1\}}^{t} \mathbb{1}\{i_s = i\}$. $\vartheta_{i,t}$ is the result of Thompson sampling from the Beta distribution.

Then we cite the same equations in Trovo et al. (2020). Use Lemma A.3, they also have the following result:

$$\sum_{t \in \mathcal{F}_\phi'} \mathbb{E}[\mathbb{1}\{i_t = i, T_{i,t,\tau} \leq \bar{n}_A\}] \leq \bar{n}_A \frac{N_\phi}{\tau}, \tag{31}$$

where $\mathcal{F}_\phi' \leq N_\phi$. Thus by choosing $\bar{n}_A = \left\lceil \frac{19}{\log\tau} \right\rceil$, we have:

$$R_A = \sum_{t \in \mathcal{F}_\phi'} \mathbb{P}\left(\vartheta_{i_\phi^*, t} \leq \mu_{i_\phi^*, t} - \sqrt{\frac{5\log\tau}{T_{i_\phi^*, t, \tau}}}\right) \tag{32}$$

$$\leq \sum_{t \in \mathcal{F}_\phi'} \mathbb{P}\left(\vartheta_{i_\phi^*, t} \leq \mu_{i_\phi^*, t} - \sqrt{\frac{5\log\tau}{T_{i_\phi^*, t, \tau}}}, T_{i_\phi^*, t, \tau} > \bar{n}_A\right) + \sum_{t \in \mathcal{F}_\phi'} \mathbb{P}\left(T_{i_\phi^*, t, \tau} \leq \bar{n}_A\right) \tag{33}$$

$$\leq \sum_{t \in \mathcal{F}_\phi'} \mathbb{P}\left(\vartheta_{i_\phi^*, t} \leq \mu_{i_\phi^*, t} - \sqrt{\frac{5\log\tau}{T_{i_\phi^*, t, \tau}}}, T_{i_\phi^*, t, \tau} > \bar{n}_A\right) + \sum_{t \in \mathcal{F}_\phi'} \mathbb{E}[\mathbb{1}\{T_{i_\phi^*, t, \tau} \leq \bar{n}_A\}] \tag{34}$$

$$\leq \sum_{t \in \mathcal{F}_\phi'} \mathbb{P}\left(\vartheta_{i_\phi^*, t} \leq \mu_{i_\phi^*, t} - \sqrt{\frac{5\log\tau}{T_{i_\phi^*, t, \tau}}}, T_{i_\phi^*, t, \tau} > \bar{n}_A\right) + \bar{n}_A \frac{N_\phi}{\tau} \tag{35}$$

The first mistake appears in the blue part. They claim that

$$\sum_{t \in \mathcal{F}_\phi'} \mathbb{E}[\mathbb{1}\{T_{i_\phi^*, t, \tau} \leq \bar{n}_A\}] \leq \bar{n}_A \frac{N_\phi}{\tau}. \tag{36}$$

This inequality is not true, as it lacks one condition $i_t = i_\phi^*$. And from the context in their proof, we know that only $i_t = i$ holds, not $i_t = i_\phi^*$.

To see why Equation (36) is wrong, consider the simple example: the algorithm always select the suboptimal arm in phase $\mathcal{F}_\phi'$. Thus, $\mathbb{1}\{T_{i_\phi^*,t,\tau} \le \bar{n}_A\} = 1, \forall t \in \mathcal{F}_\phi'$. We have

$$\sum_{t \in \mathcal{F}_\phi'} \mathbb{E}\big[\mathbb{1}\{T_{i_\phi^*,t,\tau} \le \bar{n}_A\}\big] = |\mathcal{F}_\phi'|.$$

This implies Equation (36) is not true.

In their proof, there exists other three mistakes related to Equation (36)(the numbering of the equations below is the same as in their proof):

$$\text{Eq } 43 \to \text{Eq } 44 : \sum_{t \in \mathcal{F}_\phi'} \mathbb{P}\big(T_{i_\phi^*,t,\tau} \le \bar{n}_{B^*}\big) \le \bar{n}_{B^*} \frac{N_\phi}{\tau} \tag{37}$$

$$\text{Eq } 71 \to \text{Eq } 72 : \sum_{t \in \mathcal{F}_{\Delta C,N}} \mathbb{P}\big(T_{i_\phi^*,t,\tau} \le \bar{n}_A\big) \le \bar{n}_B \left\lceil \frac{N}{\tau} \right\rceil \tag{38}$$

$$\text{Eq } 95 \to \text{Eq } 96 : \sum_{t \in \mathcal{F}_{\Delta C,N}} \mathbb{P}\big(T_{i_\phi^*,t,\tau} \le \bar{n}_{B^*}\big) \le \bar{n}_{B^*} \left\lceil \frac{N_\phi}{\tau} \right\rceil \tag{39}$$

The last two inequalities are for smoothly changes. We speculate that fixing these errors would also require proving conclusions similar to Lemma 5.3 and Lemma 5.5. However, since the Beta distribution does not have concentration properties like the Gaussian distribution (Fact 1), fixing these errors is challenging.

