# OpenReview forum: "Thompson Sampling for Non-Stationary Bandit Problems"
_TMLR — Rejected by TMLR_

### Review · Reviewer_ZJz9 · 2024-08-30

**Summary Of Contributions:**

This paper studies the non-stationary version of the standard MAB with abrupt changes.  Two versions of Thompson sampling (TS) with Gaussian prior, one with a sliding window and one with a discounting factor, are considered in this setting. The authors prove that the regret of the algorithms scale as $\tilde{O}(\sqrt{BT})$, where $B$ is the total number of switches. There is also an empirical evaluation of the algorithms which show their advantage over baselines.

TS algorithms with sliding windows or discounting factors have been proposed previously. I therefore think that the contribution of this paper is mainly the theoretical results regarding the performance of these algorithms when using a Gaussian prior.

**Audience:**

No

**Broader Impact Concerns:**

No.

**Claims And Evidence:**

Yes

**Requested Changes:**

- I encourage the authors to comment on their theoretical contributions and highlight if they think I’ve missed something.
-  Discuss how your results differ from previous work on TS for non-stationary bandits (see missing references listed in weaknesses).
- Is it possible to provide similar guarantees without knowing $B$ in advance? For example, could the size of the sliding window be tuned online?
- Could your approach be extended to the linear contextual case? This question is more out of curiosity and I believe that TS for non-stationary contextual bandits could be an interesting future extension.

**Strengths And Weaknesses:**

## Strengths

- Being able to handle non-stationarity is important for many real-world applications.

## Weaknesses
I think that the theoretical contribution of this paper is rather slim since the analysis is restricted to Gaussian priors. Most of the results seem to follow almost directly from applying results from [1]. To elaborate more on this comment:
- The first challenge listed by the authors seems to be solved by just taking a slightly larger posterior variance and adjusting the confidence variable $U$ accordingly followed by applying known results from [1]. I’m not sure that this should be viewed as a challenge.
- Lemma 5.3 seems to follow directly from the proof strategy of [2] for the stationary setting combined with Lemma A.3 [1].

There are some missing references to other papers that consider TS for non-stationary bandits and some that provide theoretical analysis:
- Thompson Sampling for Bayesian Bandits with Resets. Paolo Viappiani 2013. The 3rd International Conference on Algorithmic Decision Theory
- A Change-Detection Based Thompson Sampling Framework for Non-Stationary Bandits, Gourab Ghatak 2020. IEEE Transactions on Computers 70.10 (2020)
- Thompson Sampling for Dynamic Multi-Armed Bandits, Neha Gupta et al. 2011. 10th International Conference on Machine Learning and Applications.
- TS-GLR: an Adaptive Thompson Sampling for the Switching Multi-Armed Bandit Problem. Almi and Azizi, 2020. NeurIPS 2020 challenges of real world reinforcement learning workshop
- Nonstationary Bandit Learning via Predictive Sampling, Liu et al. 2023. Proceedings of the 26th International Conference on Artificial Intelligence and Statistics. Note: This paper explores the limitations of TS in the non-stationary setting.
- In addition,  I think it is worth highlighting that both DS-TS and SW-TS are discussed in Ch. 6.3 of the Thompson Sampling tutorial [3].


DS-TS and SW-TS require the user to tune either the forgetting parameter or the size of the sliding window. The sub-linear regret bounds of these algorithms also depend heavily on knowing the total number of switches a priori. Knowing $B$ is not possible in many real-world applications which limits the usefulness of these algorithms. In contrast, there are algorithms with similar guarantees but which don’t require the knowledge of the number of switches (see [4]).

## References
[1] On upper-confidence bound policies for switching bandit problems. Aurélien Garivier and Eric Moulines. International Conference on Algorithmic Learning Theory

[2] Further optimal regret bounds for thompson sampling. Shipra Agrawal and Navin Goyal. Artificial intelligence and statistics

[3]  A Tutorial on Thompson Sampling. Daniel J. Russo, Benjamin Van Roy, Abbas Kazerouni, Ian Osband and Zheng Wen.

[4] Adaptively Tracking the Best Bandit Arm with an Unknown Number of Distribution Changes. Peter Auer, Praktik Gajane, Ronald Ortner. 2019. 32nd Annual Conference on Learning Theory.

---

> ### Author Response · Authors · 2024-09-22
>
> Thanks for your comments.
>
> **theoretical contributions**
>  - As Reviewer pzGq mentioned, our contribution lies in providing a unified theoretical analysis method for the SW-TS and DS-TS algorithms.  In my opinion, our theoretical contribution lies mainly in some technical handling within the proof.  We use a slightly larger variance to address the first challenge. However, using a slightly larger variance is non-trivial. According to the analysis on page 19, this variance needs to be strictly greater than $\frac{2\sigma^2}{N}$ in order to ensure that $\mathbb{E}[\frac{1}{p_{i,t}}]$ has a finite upper bound. A larger variance would lead to an increase in the overall regret upper bound, so we chose $\frac{4\sigma^2}{N}$.  We use a newly defined confidence function $U$, which combined with the slightly larger variance, are crucial for the lower bound (Eq 21) in the analysis of Lemma 5.3. The second challenge actually refers to the proof of Lemma 5.3 and Lemma 5.5. In the appendix, we highlight in blue font the key techniques used that differ from previous methods. Our overall analytical framework is based on [1,2]. However, due to the errors in previous work by Trovo et al. (2020) , our work  fill a research gap in this field.
>
> **related works**
> - Thompson Sampling for Bayesian Bandits with Resets.  This paper also addresses the non-stationary problem with  TS algorithm, using a reset strategy and different priors. However, it is an experimental paper without theoretical analysis.
> - A Change-Detection Based Thompson Sampling Framework for Non-Stationary Bandits. This work uses an active algorithm, combining change-detection and TS, considering only the two-armed bandit setting.
> - Thompson Sampling for Dynamic Multi-Armed Bandits.  This article considers dynamic Thompson Sampling with a Beta prior. It is also an experimental article, lacking theoretical analysis.
> - TS-GLR: an Adaptive Thompson Sampling for the Switching Multi-Armed Bandit Problem.  This work's setting is consistent with ours, both involving arbitrary changing scenarios. Their algorithm combines TS and change detection.
> - Nonstationary Bandit Learning via Predictive Sampling  This work employs a novel sampling method-predictive sampling. They use information theory tools to proof the Bayesian regret of this method. However, we consider the frequentist regret.
> - I have read the book "A Tutorial on Thompson Sampling". They have discussed the DS-TS and SW-TS in the "Nonstationary Systems" chapter.
>
> **unknown $B$**
> - Some work (for example [4]) propose method that without knowing $B$ , but it is  difficult to analyze and implement (in my opinion). Along with the complexity of analyzing TS itself, designing and  analysing a TS algorithm that does not require knowing B is challenging.
>
> **linear contextual case**
> - There is currently work applying the TS algorithm to the linear contextual case [5]. I have read this paper. I think combining our work with their algorithm may be applied to non-stationary linear contextual problems.
>
> ## Reference
> [5] Zhang, T. (2022). Feel-good thompson sampling for contextual bandits and reinforcement learning. SIAM Journal on Mathematics of Data Science, 4(2), 834-857.

---

> > ### Comment · Reviewer_ZJz9 · 2024-10-03
> >
> > Thank you for your reply.
> >
> > I do agree with the comment by reviewer fYav regarding related work not being critically and properly commented. I think it would strengthen the paper to discuss more in depth how it relates to previous work.
> >
> > It might also be a good idea to include a table in the introduction with related work, and yours, as rows and the columns could be bounds + assumption + style of algorithm (passive/active). This would help the reader position your work in relation to previous work.

---

> > > ### Author Response · Authors · 2024-10-04
> > >
> > > Thank you.
> > > I will follow your comments to improve our paper.

---

### Review · Reviewer_pzGq · 2024-09-16

**Summary Of Contributions:**

The central point of this work is the theoretical characterization of two-variants of Thompson Sampling for non-stationary ---abruptly changing--- bandit problems:

- **Discounted Thompson sampling (DS-TS)**, that uses a discount factor $0 < \gamma < 1$ to adjust the estimate of each arm's time-varying mean reward over time
- **Sliding-window Thompson sampling (SW-TS)**, that uses a sliding window $\tau$ to adapt the estimate of each arm's mean reward over time.

The main contribution is to establish upper bounds for the expected regret of these algorithms, based on the analysis of the expected number of times a suboptimal arm is played.

The authors show in Sections 4 and 5 that the regret upper bounds of the proposed algorithms are of order $\tilde{\mathcal{O}} \left(\sqrt{TB_T}\right)$, where $T$ is the number of time steps and $B_T$ is the number of breakpoints.

The work concludes with a succinct experimental comparison of their proposed TS-based algorithms with relevant baselines for the studied problem, showcasing satisfactory performance.

**Audience:**

Yes

**Claims And Evidence:**

Yes

**Requested Changes:**

- The authors claim to "propose discounted TS (DS-TS) and sliding-window TS (SW-TS) with Gaussian priors for abruptly changing settings"
    - Do the authors agree that their algorithms, and their analysis, are actually based on uninformative/uniform priors?

    - The authors do not actually use the standard posterior update rule under uniform priors, but an increased posterior variance, which is justified theoretically.
        - Can the authors elaborate on the difficulties of the analysis of using standard Bayesian updates?
        - Would it be possible to empirically showcase whether this variance over-inflation has any empirically significant impact on performance?

- In the problem formulation of Section 3, the authors describe the non-stationary setting of their interest: i.e., abrutly changing bandit settings, where "at each breakpoint, the reward distribution changes for at least one arm."
    - The title of the work is "Thompson Sampling for Non-Stationary Bandit Problems", while the actual non-stationary setting is abruptly changing bandits
        - As a minor comment/suggestion/clarification, would the authors consider clarifying they focus on the abrutly changing setting from early on, e.g., the title or abstract?

    - In the empirical evaluation, as per Figure 2, the authors only consider the setting where the reward distribution for all arms changes synchronously, at every breakpoint.
        - Can the authors elaborate on the impact of having non-synchronous breakpoints across arms?
        - If my understanding is correct, the analysis does not require synchronous breakpoints, is that so?
        - Can the authors showcase empirically whether there is any impact of non-synchronous breakpoints?

- The theoretical analysis relies on the definition of non-stationary phases denoted as $\mathcal{S}(\gamma)$ and $\mathcal{S}(\tau)$ within Sections 5.2 and 5.3
    - Can the authors describe and elaborate on what these phases really entail?
    - The definition seems to contain the set of time-instants that are not pseudo-stationary.
        - However, in the top of Figure 1, there do not seem to be any arm-changes within $\mathcal{S}(\gamma)$, so why not consider it within the stationary phase $\mathcal{T}(\gamma)$?
    - More importantly, the abrupt arm-reward changes are determined by the "world", not the "agent".
        - Why are these pseudo-stationary and non-stationary phases $\mathcal{T}$ and $\mathcal{S}$ defined in terms of agent-specific hyperparameters $\gamma$ and $\tau$?
    - Can the authors elaborate and discuss the statement that linearly bounds "the number of elements in the set $\mathcal{S}$" in terms of $B_T \cdot D(\gamma)$ and $B_T \cdot \tau$?
        - What is exactly happening within each of these $D(\gamma)$ and $\tau$ time-instants after each breakpoint?

- In the experiments of Section 6, the authors determine the hyperparameters ($\gamma$ and $\tau$) of the presented algorithms based on the knowledge of $T$ and $B_T$ according to the theoretically justified optimal equations of Corollaries 4.2 and 4.4, respectively.
    - It would be interesting to evaluate (at least empirically) the sensitivity of the algorithms to misspecified $B_T$ and $T$.
        - E.g., how much does the algorithmic performance vary depending on under- or over-estimation of these?
    - Can the authors confirm that the algorithms do not need to know in advance $\Delta_T(i)$ and $\Delta_{max}^T$, yet these will impact their empirical performance? Can they empirically evaluate such impact?

- Based on the theoretical results of Theorems 4.1 and 4.2, one might be tempted to favor SW-TS, given the lower exponent in the $\log T$ term.
    - However, in their experimental results with Gaussian rewards of Section 6.1, DS-TS provides clearly better performance than SW-TS.
    - On the contrary, results for Bernoulli arms in Section 6.2 showcase SW-TS outperforms DS-TS empirically.
    - Can the authors discuss and explain these discrepancies?

**Strengths And Weaknesses:**

### Strenths

- The main contribution of this work is the unified theoretical characterization of the presented Thompson-sampling based algorithms for abruptly changing, non-stationary bandits
    - A unified analysis approach for regret upper bounding is presented, where both the over-estimation of suboptimal arms and the under-estimation of the optimal arm are considered.

    - The analysis (which seems correct, although I have not carefully reviewed all the proof details in the appendix) show that the regret upper bounds match the order of the lower bound proven by Garivier & Moulines (2011)
        - However, it includes an extra $\log T$ and big constant terms, that the authors argue might be a limitation of their analysis

    - Authors provide a counterexample (Appendix C) showcasing the limitations of a previous study by Trovo et al. (2020) for sliding-window Thompson sampling with Beta priors.

- The evaluation in Section 6 showcases satisfactory empirical performance of DS-TS and SW-TS, under certain information assumptions, when compared to relevant baselines.

### Weaknesses

- The work does not consider general non-stationary bandits, but those with a discrete (bounded) number of abrupt changes.
    - I do not argue this is a limitation per-se, but that the authors should state and clarify so promptly (see comments below).

- The presented algorithms are direct translations of existing techniques for UCB-type algorithms, into their TS counterparts; i.e., using discounting and sliding-window based mean estimators.

- The authors claim to propose/study TS for non-stationary settings with Gaussian priors. However, their algorithms and the analysis are actually based on uninformative priors:
    - They take the Gaussian prior's variance $\sigma_0 \rightarrow \infty$, effectively using the $\mathcal{N}(1/n \sum_{i=1}^n X_i, \sigma^2/n)$ posterior ---akin to a frequentist estimate
        - Hence, the analysis does not really characterize the effect of (non-uniform) prior knowledge ---a fully Bayesian analysis
    - There is nothing technically wrong with this prior assumption, yet one could argue that their statement "we propose the DS-TS and SW-TS with Gaussian priors for the non-stationary stochastic MAB problems" might be misleading.

---

> ### Author Response · Authors · 2024-09-22
>
> Thanks for your comments. I will respond to your concerns in the order of your questions.
>
> **priors and non-standard Bayesian updates**
> - I agree with you about the prior setting. It seems that we use uninformative priors. This is to address all $\sigma$-subGaussian reward distributions (including 1-subGaussian). See my  response (**prior with infinite variance**) to Review fYav.
> - We use a slightly large variance to perform the Bayesian updates instead of the standard Bayesian updates. This is mainly to solve the challenge under-estimation of the optimal arm, i.e., Lemma 5.3 and Lemma 5.5.  If we use the standard Bayesian updates, the lower bound (Eq 21) in the proof of Lemma 5.3 can not hold. This is crucial for our analysis. Therefore, this is mainly for the sake of theoretical analysis.  And I think we can conduct some experiments to empirically verify the impact of variance.
>
> **asynchronous and synchronous changes**
> - This paper only consider the abruptly changing settings and I will follow your comments to make clarification.
> - In my opinion, this will not affect the theory results. Since the result of Theorem 4.1 and Theorem 4.3  are aimed at each suboptimal arm, asynchronous and synchronous changes have no impact on theoretical analysis. Our experimental results indeed only considered the synchronous case; we will add experimental results for asynchronous changes later.
>
> **non-stationary phases $\mathcal{S}(\gamma)$ and $\mathcal{S}(\tau)$**
> - The definition of $\mathcal{S}(\gamma)$ and $\mathcal{S}(\tau)$ are not necessary. As you mentioned, we do this just for the convenience of representing non pseudo-stationary phase.
> - I agree with you that the abrupt changes are determined by the environment rather than the agent. However, what we are defining here is a pseudo-stationary phase, not the true stationary phase.
> $$ \mathcal{T}(\gamma)= \{ t: \forall s \in (t-D(\gamma),t],\mu_s(\cdot)=\mu_t(\cdot) \} $$
> The definition of pseudo-stationary phase includes $D(\gamma)$, so we denote it as $ \mathcal{T}(\gamma) $.
> - By the definiiton of $\mathcal{T}(\gamma)$, on the right side of any breakpoint, there will be at most $D(\gamma)$
> rounds belonging to $\mathcal{S}(\gamma)$. This means that each breakpoint brings at most $D(\gamma)$ non pseudo-stationary point, so the upper bound of $S(\gamma)$ is $B_T D(\gamma)$.
>
> **sensitivity analysis and known in advance $\Delta_T(i)$ and $\Delta_{max}^T$**
> - We also believe that conducting a sensitivity analysis on $B_T$ based on experience is not very difficult, and we will include this part of the experimental results.
> - Our algorithm does not need to know $\Delta_T(i)$. As for $\Delta_{max}^T$, it only requires T to be sufficiently large and $B_T \ll T$.  Please refer to my response to Reviewer fYav regarding the third question.
>
> **experimental performance**
> - Theoretically, the experimental performance of the SW method should be better than DS, which is consistent when the reward distribution is Bernoulli. For unbounded Gaussian rewards, I think this experimental result may be due to the posterior variance. Our posterior variance is proportional to $\frac{1}{N(\gamma)}$ or $\frac{1}{N(\tau)}$. If an optimal arm has not been selected multiple times, then $N(\gamma)$ will be very small, leading to a large sampling variance and forcing the algorithm to try selecting this optimal arm. Large sampling variance is particularly suitable for unbounded reward distributions. The minimum  of positive $N(\tau)$ in  SW method is 1. Under unbounded rewards settings, SW's exploration ability is not as good as that of the DS method.

---

### Review · Reviewer_fYav · 2024-09-19

**Summary Of Contributions:**

The paper provides an analysis of the sliding window and discounted approaches for TS in the setting of nonstationary bandits, particularly in the case where the nonstationarity is abrupt (i.e., the scenario is piecewise constant). The authors provided tight (up to logarithmic factors) upper bounds for their considered algorithms. Finally, they show the empirical performance of the algorithms over simulated environments.

**Audience:**

Yes

**Broader Impact Concerns:**

I do not foresee any significant impact of the work being of a theoretical nature, mainly.

**Claims And Evidence:**

Yes

**Requested Changes:**

See weaknesses to improve the paper accordingly

**Strengths And Weaknesses:**

Strengths:
- the paper is easy to follow
- the analysis is sound

Weaknesses:
- the literature is not analysed critically and properly commented
- unclear initialization of the algorithm
- additional assumption on DS-TS
- unclear generalization of baselines

Comments:
- maybe a more deep comment on the differences between passive and active methods in the related work would justify the choice to pick a passive approach.
- so, at the beginning of the learning period, you need to set a prior with infinite variance?
- In in the SW, how do you set the variance at the beginning of the process? Is that handled in the regret upper bound proof?

y concern is about the assumption on the discount factor, more precisely $\gamma \in (0,1)$ should satisfy:
$$\frac{\sigma^2}{\Delta_{\textit{max}}^T}(1-\gamma)^2\ln{\frac{1}{1-\gamma}}<1,$$
in fact, the original paper by Garivier Moulines (2008) did not ask for additional conditions on $\gamma$ and didn't need the knowledge of $\Delta_{\textit{max}}^T$ and is able to characterize the regret for every $\gamma \in (0,1)$, what happens to the regret when the assumption isn't met? How do we choose $\gamma$ when these parameters are not known?
- I think that a discussion on the differences and analogies with the already existing algorithms for the abruptly changing environments would give a clearer idea of the results you provided to the reader.
- I would like to have the details about this adaptation of the SW-UCB and DS-UCB algorithms since they do not have corresponding regret bounds in the original work where they were presented.
- Regarding the paragraph about the theoretical properties of the algorithms, i think it is not appropriate to put such a consideration in the experimental section


Minor:
"and she only" -> use neutral pronouns
"the Lipschitz constant of the evolve process" -> not clear
provide the explicit definition of subgaussianity
"isn't" -> is not
"i.e." -> i.e.,
"CUSUM" -> this is not the original name....cusum is the name of the change detection test.

---

> ### Author Response · Authors · 2024-09-22
>
> Thank you for reviewing the manuscript. I will respond to the issues you raised one by one.
>
> **1. passive or active methods:**  The passive methods have low computational cost and are suitable for limited resources. But their responses to environmental changes is relatively slow, making it suitable for smooth and slower-changing environments. Active methods respond more quickly to the environment but have higher implementation and computational costs.
>
> **2. prior with infinite variance：** Follow the comment in Reviewer pzGq, our method are actually based on uninformative priors. However, in my opinion, setting the initial variance to infinity is not necessary; it is merely for easier formal analysis of all $\sigma$-subGaussian reward distributions. If we design the algorithm only for 1-subGaussian distribution as in [1], then the prior variance can be set to 1, and at this point, the posterior distribution is $\mathcal{N}( \frac{\sum_{i=1}^n X_i}{n+1},\frac{1}{n+1} )$. The posterior distribution still has a simple form.
>
> **3. set the initial variance:** Our algorithm needs to know the variance of the Gaussian reward distribution, just as the UCB algorithm requires knowledge of the upper bound on rewards. Literature [2] discusses how to design bandit algorithms without using information about the reward distribution.
>
> **4. assumption on the discount factor:** This assumption about $\gamma$ is required by our analytical method. In fact, it is not necessary to know $\Delta_{max}^T$. According to our method of selecting $\gamma$,
> $$ \frac{\sigma^2}{\Delta_{max}^T}(1-\gamma)^2\log \frac{1}{1-\gamma} <  \frac{\sigma^2}{\Delta_{max}^T}\sqrt{\frac{B_T}{T\log T}} $$
> Therefore, generally speaking, if $T$ is sufficiently large and $B_T \ll T$ , this assumption holds true.
>
> We will revise our article according to your other suggestions.
>
> ## References
>
> [1] Agrawal, S., & Goyal, N. (2013, April). Further Optimal Regret Bounds For Thompson Sampling. In Sixteenth International Conference on Artificial Intelligence and Statistics (AISTATS).
>
> [2] Lattimore, T. (2017). A scale free algorithm for stochastic bandits with bounded kurtosis. Advances in Neural Information Processing Systems, 30.

---

### Decision · Action_Editor_3w1h · 2024-10-23

**Recommendation:** Reject

**Comment:**

The paper falls short at providing clear evidence to their claims. While the proofs seem correct, the lacking of rigour in defining key quantities in the proof, and justification of assumptions on the prior used by the algorithms requires addressing beyond a minor revision to the paper. The reviewers did not receive convincing answers to their remarks on this.

Another point needing addressing is a proper comparison to prior work, the reviewers pointed out a series of papers on similar settings that we believe require discussion and that have not been addressed in the current draft.

The reviewers were positive about the paper's overall results, however these shortcomings indicate the paper is not yet ready for publication and requires further revision. We then encourage the authors to address the concerns in the reviews and resubmit.

**Audience:**

Previous work is not properly discussed, the reviewers pointed out several papers proposing similar algorithms but as of yet, these papers are still not discussed in the draft.

**Claims And Evidence:**

It is my opinion that the paper falls short at providing clear evidence to their claims, many of the reviewer questions remaining unanswered even after the rebuttal. In particular:

- The paper claims to propose algorithms "with Gaussian priors for abruptly changing settings", yet both the theory and empirical results are based on an infinite variance assumption. There is substantial confusion remaining about the choice of prior in this setting.

- Key points of the proof do not have a thorough definition, for instance the quantities: $\mathcal{S}(\tau)$ and $\mathcal{S}(\gamma)$ (for which the authors' explanations have so far been unclear).

**Resubmission Of Major Revision:**

The authors may consider submitting a major revision at a later time.

---

> ### Author Response · Authors · 2024-10-26
>
> Thank you again to all the reviewers and AE. Your suggestions are very helpful to us.